# The Impact of Plant Phytochemicals on the Gut Microbiota of Humans for a Balanced Life

**DOI:** 10.3390/ijms23158124

**Published:** 2022-07-23

**Authors:** Sarusha Santhiravel, Alaa El-Din A. Bekhit, Eresha Mendis, Joe L. Jacobs, Frank R. Dunshea, Niranjan Rajapakse, Eric N. Ponnampalam

**Affiliations:** 1Postgraduate Institute of Agriculture, University of Peradeniya, Peradeniya 20400, Sri Lanka; 2Department of Biochemistry, Memorial University of Newfoundland, St. John’s, NL A1C 5S7, Canada; 3Department of Food Sciences, University of Otago, P.O. Box 56, Dunedin 9054, New Zealand; aladin.bekhit@otago.ac.nz; 4Department of Food Science and Technology, Faculty of Agriculture, University of Peradeniya, Peradeniya 20400, Sri Lanka; ereshamendis@agri.pdn.ac.lk (E.M.); niranjanp@agri.pdn.ac.lk (N.R.); 5Animal Production Sciences, Agriculture Victoria Research, Department of Jobs, Precincts and Regions, Ellinbank, VIC 3821, Australia; joe.jacobs@agriculture.vic.gov.au; 6Centre for Agricultural Innovation, School of Agriculture and Food, Faculty of Veterinary and Agricultural Sciences, The University of Melbourne, Parkville, VIC 3010, Australia; 7School of Agriculture and Food, Faculty of Veterinary and Agricultural Sciences, The University of Melbourne, Parkville, VIC 3010, Australia; fdunshea@unimelb.edu.au; 8Faculty of Biological Sciences, University of Leeds, Leeds LS2 9JT, UK; 9Animal Production Sciences, Agriculture Victoria Research, Department of Jobs, Precincts and Regions, Bundoora, VIC 3083, Australia

**Keywords:** plant foods, phytochemicals, gut microbiota, digestive process, metabolic diseases, health and wellness

## Abstract

The gastrointestinal tract of humans is a complex microbial ecosystem known as gut microbiota. The microbiota is involved in several critical physiological processes such as digestion, absorption, and related physiological functions and plays a crucial role in determining the host’s health. The habitual consumption of specific dietary components can impact beyond their nutritional benefits, altering gut microbiota diversity and function and could manipulate health. Phytochemicals are non-nutrient biologically active plant components that can modify the composition of gut microflora through selective stimulation of proliferation or inhibition of certain microbial communities in the intestine. Plants secrete these components, and they accumulate in the cell wall and cell sap compartments (body) for their development and survival. These compounds have low bioavailability and long time-retention in the intestine due to their poor absorption, resulting in beneficial impacts on gut microbiota population. Feeding diets containing phytochemicals to humans and animals may offer a path to improve the gut microbiome resulting in improved performance and/or health and wellbeing. This review discusses the effects of phytochemicals on the modulation of the gut microbiota environment and the resultant benefits to humans; however, the effect of phytochemicals on the gut microbiota of animals is also covered, in brief.

## 1. Introduction

Around 400 B.C., Hippocrates said, “death sits in the bowels” and “bad digestion is the root of all evil”, indicating the significant contribution of the human intestine to improved health [1]. Taking humans as an example, gut microbiota (GM), also called gut flora, is the name given to the microbial loads inhabiting our gastrointestinal (GI) tract, and around 100 trillion micro-organisms comprising, primarily bacteria, inhabit the human colon [2]. The human intestine is also inhabited by a lesser number of eukaryotic organisms, such as fungi, protozoa, and viruses. Anaerobic bacteria are the predominant group of the microbial community present in the human colon, which creates one of the most densely populated and diversified bacterial ecosystems in nature [3].

Although several hundred micro-organism species reside in the human colon, many relevant studies and current findings collected from the Human Microbiome Project explain that among individuals, the microbial population is greatly varied in composition [4,5]. The colonization of bacteria begins in utero, and lifelong changes take place in the composition of the GM, of which the significant alterations in number and diversity occur during the breastfeeding period and at the commencement of solid food consumption. Micro-organisms differ in number, type, and function throughout the GI tract. Still, the GM is densely populated in the large intestine where the microbiota is involved in the fermentation of undigested food components, especially fibers, some carbohydrates, and oxidized proteins, together with more associated metabolic functions [1].

In the context of herbivores, ruminants (cattle, sheep, goats, alpacas, deer, etc.) cannot directly digest plant material due to the lack of enzymes capable of breaking down high-fibrous cell wall contents, such as cellulose and hemicellulose. Rather, the GI tract of ruminants is inhabited by a large population of bacteria, protozoans, and fungi in their four-chambered stomach (rumen, reticulum, omasum, and abomasum) capable of digesting the high-fibrous plant materials. Partially or fully digested food passes through the last compartment of the stomach to the small intestine and large intestine, and further digestion and absorption of nutrients takes place by the host animal. The anatomical and functional attributes of the small intestine of ruminants is similar to humans and other animals, and ranges in length by approximately 12–30 times the animal’s body length [6].

Gut microbiota have been identified as capable sources of modern curative medicines [7]. Various details of the GM and its impacts on human health, including during childhood [8] and certain diseases, such as inflammatory bowel diseases, cardiometabolic disorders, cancer, and neuropsychiatric diseases, are considered in a large number of recent studies and reviews [9,10]. Signals that stimulate the generation of cytokines are provided by the GM, resulting in the alteration in the general development of the host immune system functions, facilitated by maturation of the immune cells [11]. The relationship between the microbiota and the host has a reciprocal nature, and it has been said that “feed your microbiota and get fed by it” [12,13].

Dysbiosis is the impairment of GM and/or their functions due to several factors, such as poor diet, insufficient exercise, stress, age, drugs, and xenobiotics [1,14]. The inter-relationship among dysbiosis and diseases, namely those related to the GI tract, such as ulcerative colitis, inflammatory bowel disease, colorectal cancer, and Crohn’s disease as well as some disorders associated with extra-intestinal metabolism, such as obesity, diabetes, cardiovascular disease, and related micro- and macrovascular complications are supported by considerable evidence [15,16]. Correspondingly, an imbalance in the microbial community can be caused by a pathological state. For example, metabolic disorders may be promoted by dysfunction of the innate immune system through the modulation of the GM [17,18].

The diet normally provides nutrients and energy for growth, development, and maintenance of life. At the same time, some food components of varying chemical structures and functionalities offer health benefits that extend beyond their nutritional value when consumed regularly, leading to improved health. These components are known as “nutraceuticals”, and foods containing these bioactive molecules are coined as “functional foods” [19,20]. Functional foods are categorized as non-nutrient (probiotics and phytochemicals) and macro- and micro-nutrients (fatty acids and vitamins) [21]. In humans, herbivores, and mice, the composition of the GM can be modified by diet [22]. Long-term dietary habits considerably influence human gut microbiota. Diet and changes in gut pH can affect several physiological aspects of the intestinal environment, particularly the absorption of nutraceuticals, micronutrients, and vitamins, consequently changing the equilibrium of the intestinal ecology [23]. Many investigators examined the effects of nutraceuticals on GM and their potential to remove pathogenic bacteria (or pathobionts) without harming the favorable bacteria (symbionts).

Phytochemicals are non-nutrient plant compounds that are biologically active and produced by the primary and secondary metabolism of plants. Phytochemicals are natural functional ingredients widely present in fruits, vegetables, seeds and nuts, whole grain products, legumes, dark chocolate, and tea, whose regular dietary intake was suggested to reduce the occurrence of many chronic illnesses [24,25]. Relatively few types of phytochemicals have been identified and isolated from plants but there are tens of thousands of phytochemicals [26]. Phytochemicals benefit the human and animal body by altering the intestinal microflora by selectively inducing the growth of some bacterial populations in the gut, referred to as “probiotics” [27]. These consist of endosymbionts, including yeast, bifidobacterium, lactic acid bacteria, and bacilli involved in human and animal GI metabolism [28].

Phytochemicals may be categorized as polyphenols, alkaloids, terpenoids (carotenoid terpenoids and non-carotenoid terpenoid), organosulfur compounds, and nitrogen-containing compounds based on their biosynthetic origins [29,30,31]. Classification of dietary phytochemicals is shown in Figure 1. Polyphenols constitute the largest group among these phytochemicals. There is increasing evidence demonstrating the ability of phytochemicals to decrease inflammation, slow the growth rate of cancer cells, reduce the formation of carcinogenic compounds, regulate gene expression and hormones’ intracellular signaling, enhance the immune system, alleviate DNA damage, decrease oxidative destruction of cells, and activate insulin receptors [32,33,34].

Compared to the micro- and macro-nutrients, the bioavailability of phytochemicals is very low within the human body due to their complex chemical structure and to their being metabolized as a xenobiotic [35]. The poor absorption of phytochemicals leads to extended retention times in the intestine where they may play a beneficial role by influencing the intestinal ecology [36]. During the last two decades, the influence of dietary phytochemicals on GI microbial population and the fundamental mechanisms assumed for their favorable effects on extra-intestinal and GI disorders have been illustrated [37,38,39]. However, there is a need to comprehensively review recent studies related to the impact of phytochemicals on gut microbiota of humans and their mechanism of action, including their relationship with metabolic diseases. This review mainly covers the effects of phytochemicals on the modulation of the gut microbiota environment and the resultant benefits to human health to establish clear directions for their potential use in human diets; however, the effect of phytochemicals on gut microbiota modulation of animals is also covered, in brief.

## 2. Human Gut Morphology and Composition of Healthy Microbiota

The surface area of the GI tract of humans is about 260–300 m^2^, which represents major interactions among many parts of the human body, interior antigens, and exterior environmental agents [40]. The intestinal microflora is a complex community consisting of 10–100 trillion microbes represented by about 1000 species. A large number of external micro-organisms and approximately sixty tonnes of food that pass through the GI tract during the average lifetime threaten the intestine’s integrity [41]. A few decades ago, the human GM was deemed to be the ‘forgotten organ’ and was simply considered to be a part of an excretion system [42]; now it is recognized as an individual organ for metabolism in the human body [43]. 

Humans and animals show a complex and mutual relationship with their symbiotic collective community of intestinal micro-organisms, namely bacteria, eukaryotes (mainly yeasts), viruses (mainly phages), archaea, and other microbial species, which have evolved through thousands of years of joint development [39]. Even though many microbial species exist, the human GI tract harbors approximately 1014 cells, chiefly anaerobes, such as *Bacteroidetes*, *Firmicutes*, Proteobacteria, and Actinobacteria [9,44]. Nearly 90% of the microbes present in the GI tract of adults are under the phyla *Firmicutes* (Gram-positive) (for example, *Lactobacillales*, *Clostridiales*, and *Ruminococcus* species) and *Bacteroidetes* (Gram-negative) (including *Bacteroides* and *Prevotella* species). Other phyla, such as Proteobacteria (Gram-negative), Actinobacteria (Gram-positive, including *Bifidobacterium*), Fusobacteria, and Verrucomicrobia (Gram-negative, including *Akkermansia muciniphila*) as well as some facultative anaerobic bacteria represent a very low proportion [2,45,46,47,48]. 

The gut microbial population is profoundly influenced by the dietary habits associated with the consumption of different types and amounts of phytochemicals [38,49,50]. Figure 2 depicts some of the known effects of phytochemicals on intestinal microflora. The gut’s microbial composition changes along the entire length of the GI tract, depending on the structure and function of the regions of the digestive system. From the proximal part of the intestine to the distal portion, a significant increase in the microbial population density and enhanced population of anaerobes occurs [44,51]. Because of the acidic environment imparted by the gastric juices (chyme) secreted by the stomach, pancreas, and biliary, the composition of the GM in the initial part of the proximal gut, mainly in the duodenum, is similar to that of the stomach [52,53].

From the duodenum to the ileum, the bacterial population increases in number and diversity toward the distal part of the gut as a result of the gradual increase in pH. This distal portion is dominated by the phyla *Bacteroidetes*, Proteobacteria (mainly *Escherichia coli*), *Firmicutes* (mainly *Lactobacillus* and *Clostridium* species), and Gram-negative facultative anaerobes [41,47]. A more favorable nutritional environment with pH in the range of 5.7–6.8 exists in the large intestine, mainly in the colon, and this induces the proliferation of the microbial population. This helps produce a more concentrated, complex, and diverse community of intestinal microflora, containing obligate anaerobes that can live at very low oxygen concentrations. The lumen of the intestine and the mucus layer of the inner lining of the intestinal tract also have different compositions and physiology of microflora. In addition, the distribution of aerobic and anaerobic microbial species also varies between them [54,55]. 

The colonization of microbes in the GI tract varies in humans and animal species. For humans, the colonization starts in the first two years of life. During gestation, the gut of infants is considered sterile, and it is not until after birth that the micro-organisms appear in the gut of infants. The first colonization of the microbes and subsequent proliferation throughout the intestine is crucial in achieving favorable outcomes by influencing the host’s immune system [56]. As the human grows and the digestive tract develops, at around 2–3 years of age, the phylogenetic diversity of intestinal microflora is established, and a stable and complex microbial ecosystem is generated [57]. 

The healthy intestinal ecology influences several vital functions (Figure 2) in the host body, including the production of bile acid, satiety, lipogenesis, digestion process, absorption of dietary nutrients, and innate immunity [58,59,60]. The GM contributes several key metabolic functions (co-metabolism) to the host body, including its ability to protect against harmful bacteria. This is carried out by maintaining and regulating the integrity and permeability of the intestinal barrier, thereby influencing the homeostasis of the host. In addition, their capacity to carry out bile biotransformation while establishing immunity of the host and fermenting carbohydrates, proteins, and lipids and synthesizing a large number of vitamins, essential and non-essential amino acids, and short-chain fatty acids (SCFAs) [61,62]. Furthermore, some dietary components such as complex oligosaccharides that are not easily digested, including a few indigestible polysaccharides (e.g., cellulose, hemicellulose, pectins, unabsorbed sugars, resistant starches, alcohols, and gums) can be metabolized by GM. This supports the bacteria in obtaining energy and nutrients for their growth and proliferation, resulting in the host regaining absorbable components from the digesta [62,63]. 

## 3. Gut Microbial Metabolism of Phytochemicals

The metabolic pathway of dietary phytochemicals in the human body is shown in Figure 3. Several reactions, including hydration, oxidation, hydroxylation, decarboxylation, methylation, dehydrogenation, glycosylation, and isomerization are involved and cause various modifications to the parent phytochemicals. The interaction of high molecular-weight phytochemicals to intestinal tissues is influenced by the metabolism of phytochemicals in food and their degradation/modification by intestinal microflora [64]. 

Considering polyphenols as an example, following absorption by the small intestine, a sequence of conjugated metabolites (water-soluble derivatives of sulfate, glucuronide, and methyl compounds) is produced in the hepatocytes and enterocytes, called Phase I. Phase I is a complex process that involves hydrolysis, oxidation, and reduction reactions. Phase II is the biological conversion of polyphenol compounds to hydrophilic metabolites through conjugation by either membrane bound or soluble cytosolic enzymes, which is less complex [65]. 

Intestinal bacteria produce metabolites, which can play several crucial functions in the body, from the unabsorbed polyphenols (around 90–95% of the total ingestion of polyphenol) by enzymatically acting on the backbone of those polyphenols remaining in the large intestine [1]. The polyphenols are metabolized by intestinal micro-organisms through glycosidic bond-splitting and heterocyclic backbone breakdown [66]. Upon absorption, the resulting metabolites of polyphenols will enter the liver via the portal vein, producing active metabolites (sulfation, methylation, and glucuronidation) by undergoing considerable degradation reactions. Further, the target tissues and cells will be exposed to these active metabolites after they are released into systematic circulation, where they can play significant physiological roles. Finally, the remaining unutilized metabolites that are in excess for the body will be excreted in urine [53]. 

## 4. Effect of Different Phytochemical Compounds on the Modulation of Gut Microbiota

### 4.1. Effect of Polyphenols

Polyphenols are chemical compounds widely distributed in plants such as vegetables, fruits, cereals, coffee, tea, and wine [67]. Dietary polyphenols are a broad group of natural heterogeneous components consisting of phenyl moieties, that are hydroxylated. Commonly polyphenols are categorized as flavonoids and non-flavonoids according to their complexity and chemical structure, including the type and number of substituting groups and the amounts of phenolic rings [68]. Examples of non-flavonoids include phenolic acids, stilbenes, curcumin, tannins, lignans, and coumarin [34,69].

Even though the most commonly found modifications are polymerization or esterification, many polyphenols are present in plants in the glycosylated form. Generally, polyphenols are received as xenobiotics in the host body after ingestion. Therefore, their bioavailability is relatively lower than macro- and micro-nutrients. Moreover, the small intestine may absorb low molecular weight compounds such as monomers and dimers due to their low complexity [70]. The compounds with a complex structure, such as oligomeric and polymeric structures, may enter the large intestine almost without any modification, degraded by microflora and absorbed by the host animal [39]. 

The actual reason behind the health benefits obtained from the intake of foods rich in polyphenols is the metabolites of polyphenol produced by GM, rather than the original components present in foods [64]. Recent studies indicated that the prebiotic properties and antimicrobial effects of dietary phenolic compounds and the antagonistic effect produced by aromatic metabolites against harmful gut microbiota, may alter the intestinal microbial ecosystem [1,39,71,72]. The evidence regarding the effect of each type of polyphenol, including flavonoids, phenolic acid, curcumin, stilbenes, lignans, and tannins, on GM is discussed below.

#### 4.1.1. Flavonoids

Flavonoids are the largest subgroup of polyphenols having more than 6000 identified compounds that have been isolated from plants [73], with the most notable being those containing pigments of flowers and plants, and play a crucial role by acting as free radical scavengers. Flavonoids can be classified into seven subgroups: flavones, flavanones, flavanols, flavonols, flavononols, isoflavones, and anthocyanins [34]. Each subgroup constitutes unique compounds that exhibit various biological activities.

##### Flavones

Many human and animal studies investigated the effects of specific flavones, including chrysin, apigenin, baicalein, and luteolin (Figure 4), on the intestinal microflora. Apigenin is particularly present at high levels in parsley (215 mg per 100 g) and celery (19 mg per 100 g). It is also found in rutabagas, tea, oranges, wheat sprouts, onions, cilantro, and chamomile [74]. The influence of apigenin on the alteration to the intestinal ecosystem is not well understood yet [75,76]. Wang et al. (2017) examined the effects of pure apigenin on the levels of both the single strain and community of human intestinal bacteria [74]. Growth profiles of the anaerobic bacteria were measured to study the influence of apigenin on the single strains of intestinal bacteria, including *Enterococcus caccae*, *Lactobacillus rhamnosus* GG, *Bifidobacterium catenulatum*, and *Bacteroides galacturonicus* as well as a fecal inoculum, were cultured by reproducing the human large intestine in vitro. The findings from that study highlighted that the growth of *E. caccae* and the microbial community cultured in vitro were effectively suppressed by apigenin compared to other examined species. It was noted that apigenin lowered the ratio between *Firmicutes* and *Bacteroidetes*.

Chrysin is widely distributed in honey and propolis [77]. Andrade et al. (2019) have examined the effect of fructose on gut microbiota and the capacity of the chrysin to influence the observed putative variations in an in vivo study [78]. They found that chrysin did not modify the composition of the intestinal ecology, but the abundance of *E. coli* and *Lactobacillus* were considerably enhanced by fructose, while the ratio between *Firmicutes* and *Bacteroidetes* was increased in rats treated with fructose and chrysin. This study was the first to report that chrysin can interfere with the impacts of fructose on the intestinal ecosystem, which may be responsible for the features of metabolic syndrome induced by fructose.

Baicalein has several pharmacological effects, including anti-inflammatory, antioxidant, and antitumor properties. Gao et al. (2018) reported that treatment with baicalein significantly modified the population density of six bacterial genera in senescence-accelerated mouse prone 8 (SAMP8) mice by analyzing their intestinal microflora [79]. In another study, Zhang et al. (2018) treated male Wistar rats with baicalein for four weeks and examined the ability of baicalein to manipulate the intestinal microflora. It was observed that the ratio of *Bacteroidetes* to *Firmicutes* was significantly decreased, the relative abundance of *Bacteroidales*, Bacteroidaceae, Porphyromonadaceae, and Verrucomicrobiaceae was significantly increased, while that of Streptococcaceae, Desulfarculaceae Deferribacteraceae, and Ruminococcaceae was reduced in baicalein-treated rats [80].

##### Flavanones

Citrus fruits are the richest source of dietary flavanones for humans, while aromatic herbs (mint) and tomatoes contain flavanones, but in lower percentages [81]. Although flavanones are found in limited amounts in the diet, they are also one of the main dietary flavonoids since citrus fruits and juices are frequently consumed worldwide. Naringenin and hesperetin (Figure 5) are the most abundant flavanones present in grapefruit and sweet oranges, respectively, and are the primary reason for their specific bitter taste [82]. 

Previous studies have demonstrated that through suppressing the growth of the harmful microbes and activating certain beneficial microbes, flavanones of citrus could alter the function and composition of the intestinal ecosystem [83]. Homeostasis of the intestine could be maintained by ingesting citrus flavanones, which might explain the mechanism of their beneficial health effects [84]. 

The primary target of the studies examining the influence of citrus flavanones or citrus fruit-based food products on the modulation of gut microflora is their capacity to promote the growth of beneficial microbes (such as the species of *Lactobacillus* and *Bifidobacterium*), to decrease the population of harmful bacteria, and to trigger generation of short-chain fatty acids (SCFAs). The findings of an in vitro study [85] stated that, even though the parent compounds failed to influence microbial species, the incubation of both naringenin and hesperetin (aglycones of citrus flavanone) decreased the population of several bacterial species after 24 h of incubation, which included the abundance of *Enterococcus caccae*, *Bifidobacterium catenulatum*, *Ruminococcus gauvreauii*, *Bacteroides galacturonicus*, and *E. coli*. Only naringenin suppressed the growth of *Lactobacillus* spp., and it showed the inhibitory action at a minimum level of 250 µg/mL [85]. These findings are in agreement with the results reported by Parkar et al. (2008) who also reported that naringenin can suppress the proliferation of *Salmonella typhimurium*, *E. coli*, *Lactobacillus rhamnosus*, and *Staphylococcus aureus* at a minimum inhibition concentration (MIC) of 62.5 µg/mL for *Staphylococcus aureus* and at 125 µg/mL for the other three bacteria [86].

Hesperidin (Figure 5), which is formed by the conjugation of hesperetin with rutinose, is the principal flavonoid widely distributed in citrus fruits [87]. Once the hesperidin enters the colon, the connected rutinose groups are split by GM, producing hesperetin for further absorption in the large intestine [88]. A recent study reported that oral supplementation of hesperidin at two dose levels of 100 and 200 mg/kg for four weeks (three times per week) significantly altered the intestinal microbiota composition in rats [89]. The population of *Staphylococcus* was increased and the proportion of *Clostridium coccoides* and *Eubacterium rectale* was decreased by both hesperidin dosages, while the abundance of *Lactobacillus* was enhanced by the high dose treatment of hesperidin. 

Previous studies conducted in humans examined the influence of hesperidin administration on the composition of the intestinal ecosystem and the production of SCFAs [84,90,91]. The ratio between butyrate and total SCFA was increased by the daily oral administration of 500 mg of a citrus extract (containing hesperidin of >80% and naringin of >4%) for 12 weeks in a randomized, placebo-controlled study of healthy individuals showing features of metabolic disorders, while there was no change observed in the absolute concentrations of SCFAs in the fecal samples [84,92].

Lima et al. (2019) showed that continuous consumption of orange juice for two months could notably enhance the proportion of total fecal anaerobes and *Lactobacillus* spp. in healthy volunteers [90]. However, the study did not state the amount of flavanone. Additionally, a considerable decline in the concentration of ammonium and a rise in the ratio of acetate to total SCFA were observed compared to the control treatment. In healthy volunteers, modulation of the composition of the microbial community was observed with the daily ingestion of two orange juices that had different flavanone concentrations for seven days [91]. The findings showed that the most significant increase was observed in the population belonging to the functional taxonomic unit of clostridia from the families of Veillonellaceae, Ruminococcaceae, Odoribacteraceae, Tissierellaceae, and Mogibacteriaceae [91].

Briefly, the findings from several human, animal, and in vitro experiments indicate that supplementation of citrus flavanone can shift the intestinal ecology composition or proliferation of a particular taxon. Even though many studies have repeatedly depicted the suppression of the growth of Enterobacteriaceae by citrus flavanone, the results of some other research vary. Unfortunately, there are no studies involving the analysis of fecal metabolites of humans to investigate the influence of flavanones on the intestinal microflora [84].

##### Flavanols

The flavanol subgroup is primarily composed of catechins, which are present in higher concentrations in the fruit’s skin as opposed to the pulp of the fruit. Catechins are found at high concentrations in green tea and are responsible for several health benefits of green tea. Although there are several compounds included in this group, only epicatechin, epigallocatechin, epicatechin gallate, and epigallocatechin gallate (Figure 6) are reported to affect the intestinal community [31].

Cueva et al. (2013) examined in vitro fermentation of bacterial loads by flavan-3-ol using two purified fractions from grape seed extract (GSE): GSE-M (70% monomers and 28% procyanidins) and GSE-O (21% monomers and 78% procyanidins) and observed, using a fluorescent in situ hybridization technique, the appearance of the metabolites of phenols. During the fermentation, both grape seed flavanol fractions increased the proportion of *Enterococcus* and *Lactobacillus* and reduced the proportion of *C. histolyticum*. Overall, the results revealed that flavan-3-ol can stimulate changes that could influence their potential bioactivity and bioavailability by modifying the composition of colonic microflora and intrinsic catabolic activity [93].

Tzosunis et al. (2008) used a batch-culture fermentation system where pH is controlled modeling the distal portion of the large intestine colon to evaluate the bidirectional metabolic relationship among intestinal microflora and (+)-catechin and (−)-epicatechin. Remarkably, the proliferation of certain bacterial loads was affected by (+)-catechin, where the growth of *Bifidobacterium* spp., *E. coli*, and a group of *C. coccoides* and *E. rectale* was considerably enhanced, at the same time, the proportion of the *C. histolyticum* bacteria was decreased, leading to the beneficial effects in the body. Despite this, (−)-epicatechin significantly promoted only the proliferation of the group of *C. coccoides* and *E. rectale*, proving their poor activity. Both (+)-catechin and (−)-epicatechin showed noticeable potential prebiotic properties at a minimum concentration of 150 mg/L. The considerable change in the specific populations of gut microbiota resulted from the incubation of (+)-catechin could be related to the transformation of (+)-catechin to (−)-epicatechin by bacteria since the same metabolites of 5-(3′, 4′-dihydroxyphenyl)-γ-valerolactone, 5-phenyl-γ-valerolactone and phenyl-propionic acid are produced by the transformation of both (−)-epicatechin and (+)-catechin. In conclusion, these results suggest that the intake of diets rich in flavanol may contribute to intestinal health by acting as prebiotics [94].

The ability of (−)-epigallocatechin gallate (EGCG) to influence the modification of intestinal microflora and the output of fermentation in the large intestine was investigated by Unno et al. (2014) who found that the growth of specific microbial species such as *Clostridium* spp., *Bacteroides*, *Bifidobacterium*, and *Prevotella* in the colon was affected by dietary EGCG in rats [95]. Further, Cheng et al. (2017) investigated the influence of (−)-epigallocatechin 3-*O*-(3-*O*-methyl) gallate (EGCG3″Me) in a human flora-associated mice model with obesity induced by a high-fat diet. After 8 weeks of supplementation of EGCG3″Me, the proportion of *Bacteroidetes* was dramatically increased, while the population of *Firmicutes* had declined. In this study, it was found that EGCG3″Me can act as a prebiotic and potentially have therapeutic effects in modulating the colonic ecosystem, which assists in gut dysbiosis inhibition [96]. In a study by Lee et al. (2006), the growth of the *E. coli*, *C. coccoides*, and *E. rectale* was promoted, while the growth *C. histolyticum* was suppressed by (+)-catechin. Moreover, (+)-catechin did not affect the population of beneficial microflora, namely *Lactobacillus* spp. and *Bifidobacterium* spp. [97]. Preclinical data of an in vitro study in a bacterial medium revealed that epicatechin gallate sensitizes methicillin-resistant β-lactam antibiotics *S. aureus* to become sensitive to β-lactam antibiotics [98].

##### Flavonols

The flavonols subgroup constitutes different compounds, including quercetin, kaempferol, myricetin, rutin, and isorhamnetin (Figure 7). Generally, only quercetin and kaempferol appear to have been the focus of many studies investigating their ability to manipulate the GM. Fruits and vegetables contain quercetin in high amounts, and the approximate daily intake of quercetin in a typical Western diet was estimated to range from 0 to 30 mg/day, depending on the intake of fruits and vegetables [99]. Foods such as berries, apples, onions, and kale are considered the richest sources of quercetin [100]. Recent studies reported that the abundance of beneficial *Lactobacillus*, *Bacteroides*, Clostridia, and *Bifidobacterium* was significantly increased, and the proportion of *Enterococcus* and *Fusobacterium* was decreased by the administration of quercetin, which modified the composition of the intestinal ecosystem [101,102]. In an earlier study, single-molecule RNA sequencing was used along with Helicos technology to investigate the influence of quercetin supplementation on the commensal colonic bacteria, including *Enterococcus caccae*, *Ruminococcus gauvreauii*, and *Bifidobacterium catenulatum*, by analyzing the gene expression profiles and examining the cell morphology and growth patterns between groups treated with and without quercetin [103]. It was found that phenotypically, quercetin moderately suppressed the growth of *E. caccae*; mildly inhibited the growth of *B. catenulatum;* and did not affect the growth of *R. gauvreauii* [103].

Etxeberria et al. (2015) examined the consequences of the supplementation of quercetin and *trans*-resveratrol on a high-fat sucrose diet (HFS)-induced dysbiosis of gut microbiota. In this study, Wistar rats were randomly divided into four groups, namely those receiving an HFS diet enriched with or without quercetin (30 mg/kg BW/day), *trans*-resveratrol (15 mg/kg BW/day), or a mixture of the two components (quercetin and *trans*-resveratrol) at the same dosages used in the other treatments. This study revealed that the administration of quercetin resulted in a significant effect on the composition of GM at various levels of taxonomy, reducing the ratio between *Firmicutes* and *Bacteroidetes* and suppressing the multiplication of gut microbes linked to diet-induced obesity, including *E. cylindroides*, Erysipelotrichaceae, and *Bacillus*. In summary, dysbiosis of gut microbiota induced by the HFS diet was effectively attenuated by the supplementation of quercetin. However, there was no significant influence on the GM composition by the supplementation of *trans*-resveratrol alone or in combination with quercetin. Moreover, the individual effect of quercetin was reduced when combined with *trans*-resveratrol [104].

Numerous medicinal and edible plants and herbs contain kaempferol, which is regarded as an important compound in the flavonol subgroup. Kaempferol has several pharmacological effects, including its ability to act as an antioxidant and anti-inflammatory agent [105]. A study observed that the colonic ecosystem and its metabolism were regulated by a high concentration of kaempferol in the large intestine [106]. Moreover, Kawabata et al. (2013) conducted an in vitro study to investigate the growth of *B. adolescentis* treated with flavonols by incubating the *B. adolescentis* obtained from the human colon with various flavonols, including quercetin, kaempferol, fisetin, myricetin, and galangin under anaerobic conditions. Quercetin and fisetin did not affect or mildly affected the growth rate (inhibited by 20% after 6 h of treatment). In comparison, galangin showed about 30% to 70% suppression in the growth rate of *B. adolescentis* when incubated for 1 to 6 h. This study suggested that, except for galangin, other tested flavonols exhibited no or mild anti-bacterial properties against *B. adolescentis*, which is considered beneficial for gut health [107].

An in vitro experiment was carried out by Duda Chodak (2012) to evaluate the effects of rutin and quercetin on certain colonic microbial species. Rutin was used at rates of 20, 100, and 250 µg/mL, and quercetin was examined at 4, 20, and 50 µg/mL against six inoculate bacteria species (*B. galacturonicus*, *B. catenulatum*, *Ruminococcus gauvreauii*, *Lactobacillus* sp., *E. coli*, and *E. caccae*). The obtained results revealed that, although the inhibitory action of rutin against bacterial proliferation was weak, quercetin had a significant inhibitory activity, which depended on the concentration used against the examined bacterial species [85].

##### Flavanonols

The flavanonols subgroup constitutes astilbin, engeletin, and taxifolin (Figure 8). Generally, intestinal microflora metabolizes these compounds and the flavanonols have anti-inflammatory properties [108,109]. There are no studies related to the effects of these compounds on the intestinal microbiota.

##### Isoflavones

Isoflavones, one of the flavonoids subgroups, contain different compounds, including daidzin, daidzein, formononetin, glycitein, and genistein (Figure 9). The chemical structures of isoflavones resemble the structure of 17-β-estradiol, an estrogen steroid hormone. Isoflavones exhibit estrogenic activity and are naturally present in many plants, among which soy is considered one of the most abundant sources [110]. Generally, isoflavones present as conjugates of isoflavone glycoside (such as genistin, daidzin, and glycitin) in unfermented soy foods and soy milk, which have low estrogenic effects and bioavailability. Isoflavone aglycones (genistein, daidzein, and glycitein) should be liberated from the corresponding glycosides to be absorbed and attain a complete activity [111]. The cellular *β*-glucosidases and *β*-glucosidases from the components of intestinal microflora are responsible for the release of isoflavone aglycones [112]. Cellular enzymes and other constituents of the colonic microflora can further hydrolyze and metabolize isoflavone aglycones producing more active compounds (for example, equol from daidzein) or inactive metabolites [113].

Soy and soy products are the major sources of dietary isoflavones. Factors that affect the total quantities of genistein and daidzein (including glycosides) present in soy products are the type of cultivar, preparation method, and extent of ripeness [114]. Frequent consumption of soy and soy products in Asian populations resulted in the greater prevalence of equol-producing microbes within the colon [115]. Another study revealed that the intake of dietary daidzein significantly increased the population of two equol-generating bacteria, *Slackia isoflavoniconvertens* and *Asaccharobacter celatus* [116]. Moreover, numerous studies conducted in humans and rodents showed that consumption of soy or soy food products affects intestinal ecology [114,117,118,119,120]. It was also stated that the total amount and/or relative percentage of the particular microbial ecosystem in the intestine might be altered by genistein in food or supplemental forms [121].

A study showed that after one week of soy isoflavone supplementation, colonic microflora composition was significantly modified in 17 postmenopausal women receiving a soy bar that contained 1 g of saponin and 160 mg of soy isoflavones (including daidzein, genistein, and glycitein) [122]. It was found that after the administration of soy isoflavones, all subjects showed a significant increase in the population of bifidobacteria in the colonic microflora while reducing that of lactobacilli, and in equol-generating subjects, the growth of bifidobacteria and eubacteria were greatly promoted than in non-producers [122]. Another study showed similar results where soy isoflavones (100 mg per day) were given to 39 postmenopausal women for 2 months. In this study, isoflavone supplementation enhanced the proportion of *Faecalibacterecterium*, *Eubacterium*, *Clostridium*, *Lactobacillus*, *Enterococcus*, and *Bifidobacterium*. At the same time, the levels of *Clostridium* and *Eubacterium* were significantly increased in equol producers (*n* = 12) [123]. When cows were used as a model, supplementation with soy isoflavones also increased the population of *Firmicutes* [124]. However, no modification in the composition of gut microbiota was observed in an experiment involving 16 menopausal women receiving soy isoflavone (80 mg per day) [125] suggesting that soy isoflavones may have a subject-dependent and/or dose-dependent influence on the intestinal ecology.

Although different studies revealed varying results, animal and human studies suggest that the ingestion of soy-based foods promoted lactobacilli and bifidobacteria growth, and the ratio of *Firmicutes* to *Bacteroidetes* can be changed [126]. That said, a reciprocal relationship between the soy isoflavones and colonic microflora in studies in healthy volunteers is difficult to establish. Therefore, further research is required to explain the effect of soy isoflavones on the colonic microflora structure involving gnotobiotic animal models transplanted with microflora obtained from various donors [127].

##### Anthocyanins

Anthocyanins belong to the flavonoids group of polyphenols. They are plant pigments imparting deep red/purple/blue colors in plant-derived food products [128]. Although anthocyanins are naturally present in the form of aglycones and glycosides of flavylium (2-phenyl benzopyrylium) salts, they are different from the structure of the salts [129]. It is reported that approximately 700 anthocyanin compounds have been identified and isolated from plants, but only 6 anthocyanidins are widely studied, namely delphinidin, cyanidin, pelargonidin, malvidin, petunidin, and peonidin (Figure 10) [130]. 

The majority of ingested anthocyanins are biologically converted to their metabolites by the gut microbiota and are absorbed in the large intestine since they are not utilized in the upper part of the GI tract. Intestinal microbiota can metabolize anthocyanins, and in turn, anthocyanins and/or their metabolites can alter the composition of the gut ecosystem by altering the abundance of particular bacteria [131]. Under in vitro conditions and in humans, anthocyanins can increase the prevalence of favorable bacteria, including *Lactobacillus*-*Enterococcus* spp. and *Bifidobaterium* spp. [132,133]. Similarly, Sun et al. (2018) revealed that the growth of *Bifidobacterium infantis*, *Bifidobacterium bifidum*, *Lactobacillus acidophilus*, and *Bifidobacterium adolescentis* could be stimulated and the proliferation of harmful *S. typhimurium* and *S. aureus* were suppressed by purple sweet potato anthocyanins and monomers of peonidin-derived anthocyanin in in vitro microbial cultivations [134]. In a study by Chen et al. (2018) [135], it was observed that the prevalence of beneficial *Faecalibacterium prausnitzii*, *Eubacterium rectale*, and *Lactobacillus* was increased and the abundance of *Desulfovibrio* spp. and *Enterococcus* spp. was decreased by the supplement of black raspberry anthocyanins. Similar findings were reported by Zhu et al. (2018), who reported that anthocyanin of black rice and cyanidine-3-*O*-glucoside considerably stimulated the growth of lactobacilli and bifidobacteria genus [136].

In an earlier study, malvidin-3-*O*-glucoside was incubated with fecal slurry under in vitro conditions. The results indicated that malvidin-3-*O*-glucoside increased the population of total bacteria, such as *Lactobacillus* spp. and *Bifidobaterium* spp., while the abundance of *Bacteroides* spp. was unaffected [132]. This stimulating effect of malvidin-3-*O*-glucoside on the proliferation of beneficial microbes can be improved by mixing malvidin-3-*O*-glucoside with other anthocyanins. In an in vivo study that analyzed the fresh fecal samples of eight healthy volunteers (25–30 years), Zhang et al. (2016) reported that anthocyanins of purple sweet potato increased the population of *Lactobacillus*-*Enterococcus* spp. and *Bifidobacterium*, while *Clostridium histolyticum* and *Bacteroides*-*Prevotella* were decreased, although total bacteria count was not affected by the treatment [137].

Red wine extract was incubated with bacteria isolated from human feces, and the results indicated that the abundance of *C. histolyticum* was decreased with no other observed changes [138]. While this does not reflect the pathway expected during digestion, another randomized, crossover-controlled intervention study has also shown a similar observation concerning the abundance of *C. histolyticum* after consuming red wine in humans [139]. It is likely that anthocyanins were not the only compounds responsible for the above effect found with red wine consumption since red wine contains a complex mixture of polyphenols such as flavanols, flavonols, anthocyanins, phenolic acids, etc. [138]. Further, Vendrame et al. (2011) showed that the population of *Bifidobaterium* spp. was significantly enhanced in human volunteers after consuming a blueberry drink for 6 weeks [140].

A study conducted on polygenic obese mouse models fed with diets supplemented by six types of berries containing different profiles of anthocyanin for 12 weeks showed that the proportions of *Actinobacteria* and obligate anaerobic bacteria were significantly increased in the intestine by the treatment [141]. Later, the influence of a mixture of prebiotics and anthocyanin on the colonic ecosystem and colonic inflammation was determined using uncomplicated obese male and female volunteers in an open-label study for 8 weeks. The frequent intake of the mixture of prebiotics and anthocyanins resulted in a favorable alteration in the intestinal community of microflora [142]. 

Overall, in vitro animal and human intervention studies suggest that anthocyanins can promote the proliferation of beneficial bacteria, including *Bifidobacterium* spp. and *Lactobacillus* spp. These species contribute beneficial roles in the colon, such as the antimicrobial action against harmful bacteria via competing for adhesion sites and growth substrate, and the production of SCFAs. Furthermore, a decrease in the abundance of pathogenic bacteria, such as *C. histolyticum*, has been observed, which is responsible for inflammatory bowel disease and tumor-inducing actions [132,143]. Further studies are required to understand the overall impact of these compounds and define the mechanism of action that anthocyanins exert on the community of intestinal microbiota. This can be attained by performing well-designed clinical trials in humans and animals with a balanced experiment design of the subjects’ age and genetic backgrounds. This should cover different forms and dosages of anthocyanins and establish a consistent approach to control diets [128,144]. 

#### 4.1.2. Curcumin

Curcumin (Figure 11) belongs to the subgroup of polyphenols and the rhizome part of the *Curcuma longa* is considered the richest source of curcumin, which is used for cooking and conventional medicine [31]. Typically, it is extracted from the turmeric rhizome by solvent extraction and refinement of the extract is by crystallization. Numerous animal studies have reported that the intestinal microbial community may be influenced by curcumin. For instance, a deficiency of estrogen causes detrimental changes in the colonic microbiota, and administration of curcumin can partly re-establish the composition of the typical GM in ovariectomized rats [145]. Ohno et al. (2017) revealed that the prevalence of bacteria-generating butyrate and level of fecal butyrate can be increased by curcumin nanoparticles at a dosage of 0.2% (*w*/*w*). Further, curcumin nanoparticles stimulate the NF-κB in epithelial cells of the intestine and the inhibition of mucosal mRNA expression of inflammatory mediators [146]. 

In a colitis mouse model, curcumin enhanced the relative proportion of *Lactobacillales*, while reducing that of *Coriobacterales* [147]. Curcumin may alter intestinal barrier function by remaining in the colonic mucosa in higher concentrations, thereby minimizing inflammation caused by circulating lipopolysaccharide producing bacteria. Feng et al. (2017) reported that the administration of curcumin restores the gut barrier function in high-fat-diet-fed rats and alters the intestinal microflora composition and diversity [148]. Higher proportions of anti-inflammatory lactobacilli and bifidobacteria and fewer loads of pro-inflammatory *enterococci* and *enterobacteria* were observed in animals supplemented with curcumin [149].

In a double-blind, randomized, placebo-controlled pilot study, the influence of supplementation of dietary curcumin and turmeric on the human intestinal ecosystem was evaluated [150]. It was found that, with time, colonic microflora showed a notable and individualized modification. The relative prevalence of 71 and 56 taxa were significantly reduced by the supplementation of turmeric and curcumin, respectively. These studies reveal that proliferation, growth, or the existence of beneficial microbes in the intestinal community might be promoted by curcumin.

#### 4.1.3. Phenolic Acids

Phenolic acids (aromatic acids) are the second major subgroup of polyphenols, consisting of a phenyl ring and a carboxylic group [31], and are produced by the shikimate pathway. Although phenolic acids are present in numerous food products, they are widely distributed in berries, wine, and whole grains. Phenolic acids are divided into two main groups, namely hydroxybenzoic acids and hydroxycinnamic acids [151]. Phenolic acids have enormous health benefits.

##### Hydroxybenzoic Acid

Hydroxybenzoic acid is a phenolic derivative of benzoic acid, which can be obtained both naturally and synthetically. It contributes to several derivatives such as gallic acid, protocatechuic acid, syringic acid, vanillic acid, and *p*-hydroxy-benzoic acid (Figure 12) [151]. Protocatechuic acid (3, 4-dihydroxybenzoic acid) is abundant in human diets through foods such as white grapes, bran, brown rice, gooseberries, olive oil, onions, plums, and almonds [152]. It has been reported using a rodent model that altered gut microbial communities can be repaired by an n-butanol fraction of *Trianthema portulacastrum* rich in protocatechuic acid that lowered the relative abundance of inflammatory microbes such as *Helicobacter*, *Mucispirillum*, and Lachnospiraceae [153]. Wang et al. (2019) examined the influence of protocatechuic acid on the gut health of Chinese yellow-feathered broilers. The results revealed that microflora diversity was altered by dietary protocatechuic acid. In comparison to control group broilers, protocatechuic acid-treated broilers had more *Firmicutes* and Actinobacteria that are beneficial to gut health and fewer Proteobacteria and *Bacteroidetes* that have pro-inflammatory effects [154].

##### Hydroxycinnamic Acid

Hydroxycinnamic acids (HCAs) are the most abundant phenolic acids in plants [155]. The major factor contributing to this is that HCAs are attached to the cell walls of plants [156]. Many enzymes needed for the hydrolysis and metabolism of the complex structure of this polyphenolic compound are not present in the human genome [157]. However, these glycans can be fermented by the anaerobic intestinal microflora, where different micro-organisms liberate HCAs in the human intestine, thus considerably impacting human health [59,158]. Hydroxycinnamic acids contain a wide range of compounds, such as caffeic acid, *p*-coumaric acid, sinapic acid, chlorogenic acid, and ferulic acid (Figure 13) which are all recognized to manipulate the intestinal ecosystem [159].

Oral administration of 1 mmol/L of caffeic acid for 15 days enhanced the abundance of mucin-deteriorating *Akkermansia* and could restore the abundance of gut microbiota and suppress the rise in the ratio between *Firmicutes* and *Bacteroidetes* in female C57BL/6 mice model with colitis induced by dextran sulfate sodium (DSS) [160]. In an in vitro study analyzing the fecal microbiota of humans, Mills et al. (2015) showed that after 10 h of exposure to chlorogenic acid at a concentration of 80.8 mg, colonic microbiota was selectively modulated via increasing the abundance of *Bifidobacterium* spp., *E. rectale*, and *C. coccoides* [161]. Oral administration of 150 mg of chlorogenic acid for 6 weeks in ICR male mice with dysbiosis induced by a high-fat diet enhanced the proliferation of Bacteroidaceae and Lactobacillaceae, while suppressing that of Erysipelotrichaceae, Lachnospiraceae, Ruminococcaceae, and Desulfovibrionaceae, indicating restoration of normal intestinal microbiota [162]. 

Ma et al. (2019) used male apolipoprotein E (ApoE−/−) mice with non-alcoholic fatty liver disease induced by a high-fat diet to investigate the influence of oral administration of ferulic acid (30 mg/kg). This study suggested that the colonic microflora composition was modified by decreasing the indole-3-acetic acid secretion and changing the ratio between *Firmicutes* and *Bacteroidetes* [163]. Yang et al. (2019) investigated the influence of oral administration of 200 mg/kg sinapic acid for 8 weeks in 30-week-old male Wistar rats fed a high-fat diet (45% fat) [164]. Gut microbiota diversity was improved by enhancing the population of *Dorea* and *Blautia* of the Lachaospiraceae family while decreasing that of Desulfovibrionaceae and *Bacteroides*, which generally contribute to human diseases and inflammation [165,166]. 

#### 4.1.4. Stilbenes

Stilbenes are widely distributed in red grapes, certain berries, peanuts, and many other plants. Stilbenes are generally produced by the phenylpropanoid pathway, and one of their important features is the presence of aromatic rings connected to an ethane bridge [167]. Stilbenes comprise different compounds such as resveratrol, piceatannol (Figure 14), pinostilbene, batatasin III, oxyresveratrol, and thunalbene. Jaimes et al. (2019) investigated the influence of six stilbenoids on the composition of the intestinal microflora, and the results revealed that the analyzed stilbenoids modify the GM as observed in a human gut model of fecal fermentation at physiological levels of 10 µg/mL. The ratio between *Firmicutes* and *Bacteroidetes* was significantly lowered, and the responses from different strains from the Lachnospiraceae family and the relative proportion of strains from the *Clostridium* genus showed a consistent reduction. Among the stilbenoids groups studied, resveratrol and piceatannol highly contributed to the observed responses, followed by thunalbene and batatasin III [168].

##### Resveratrol

Resveratrol, also known as 3, 5, 4′-trihydroxystilbene, is a widely distributed polyphenol subgroup present in different plants. Although grapes, red wine, and peanuts contain considerable amounts of resveratrol, *Polygonum cuspidatum* is the richest natural plant source of the compound [169]. Resveratrol is also available commercially as tablets and is used as a nutritional supplement [170]. In recent years, studies on resveratrol have significantly increased due to its wide range of biological activities such as antioxidant, anti-diabetes, anti-obesity, and improvement in colonic micro-organisms [171,172]. Numerous studies revealed that resveratrol can reduce body fat and body weight while improving obesity and glucose homeostasis parameters by stimulating alterations in the colonic microflora. Other studies have shown that treatment with 0.4% resveratrol could enhance the population of Parabacteroides and Bacteroides while reducing those of *Akkermansia*, Lachnospiraceae, *Moryella*, and Turicibacteraceae in C57Bl/6N mice [173,174]. 

It has been suggested that resveratrol could improve atherosclerosis by suppressing the generation of trimethylamine in the gut, which is a trimethylamine-N-oxide (TMAO) precursor, by simulating intestinal microflora in mice. Fecal excretion and de-conjugation of bile acid are also increased by resveratrol via enhancement of the abundance of colonic bacteria with bile salt hydrolase activity, such as *Bifidobacterium* and *Lactobacillus* [175]. Moreover, the ratio of *Firmicutes* to Proteobacteria was increased by resveratrol (50 mg/L) examined in Sprague Dawley rats [176]. Most et al. (2017) conducted a study supplementing a combination of resveratrol and epigallocatechin-3-gallate in humans for 12 weeks. It was observed that the prevalence of *Bacteroidetes* was significantly decreased and the proportion of *Faecalibacterium prausnitzii* tended to reduce in overweight men [177]. Moreover, resveratrol promoted the multiplication of *Lactobacillus* and *Bifidobacterium* and prevented the virulence factors of *Proteus mirabilis* [178].

##### Piceatannol

Several groups of plants, particularly white tea and grapes, contain piceatannol, which is a hydroxylated analog of resveratrol [31]. Setoguchi et al. (2014) indicated that metabolic stability of piceatannol is greater in comparison to resveratrol [179]. Piceatannol is distributed at nearly equal concentrations as resveratrol in edible plants, fruits, and red wines [180]. Using C57BL/6 mice fed with a high-fat diet, the influence of piceatannol on gut microbiota was evaluated. It was observed that gut microbiota composition was significantly altered in piceatannol-treated animals and the modulation of intestinal microflora stimulated by the high-fat diet was restored by piceatannol, via significantly reducing the abundance of *Firmicutes* that are unfavorable and increasing those of *Bacteroidetes* that are favorable to enhanced gut health [181].

#### 4.1.5. Lignans

Dietary lignans are a group of phytoestrogens found as aglycones or glycosides in plants [182]. Different parts of about 70 diverse species of plants, such as roots, stems, rhizomes, leaves, fruits, and seeds contain a significant amount of lignans. Specifically, oilseeds, mainly flaxseeds and grains with bran, are the richest sources [183]. Although estrogen-like activities are not found in the dietary lignans themselves, the colonic microbial ecosystem metabolizes them to produce enterolignans (or mammalian lignans), including enterolactone (EL) and enterodiol (ED). The prevalence of *Ruminococcus* species, such as *R. lactaris* and *R. bromii* [184], and the abundance of *Methanobrevibacter* [185] and *Lactobacillus*-*Enterococcus* [186] are associated with the production of EL.

Recently, Corona et al. (2020) evaluated how the composition of the colonic microbiota of both younger and premenopausal females is affected by lignans present in oilseed mix and it was observed that oilseeds rich in lignans have a strong impact on the fecal microbiota, generating a varying profile of enterolignan. Further research is required to investigate the long-term impacts of diets rich in lignan on the intestinal microflora and to discover ways to increase the abundance of bacterial species producing enterolactone. Plants and plant products containing lignans consumed by animals and humans have many advantages, not only improving the GM population and gut health but also providing a range of health benefits. Lignans can improve the antioxidant status of both tissue systems and the whole body, prevent cancer by limiting the proliferation of cells via anticarcinogenic effects, improve the immune status of individuals by providing defense against infectious diseases, and act as pro-inflammatory compounds in reducing arthritis and obesity [187].

#### 4.1.6. Tannins

Tannins are a subgroup of polyphenols which comprise several components of different molecular weights that are widely distributed in nature [188]. Proteins can be precipitated by tannins. Depending on the molecular structure, tannins are divided into hydrolyzable tannins (HTs) and condensed tannins (CTs). An example of hydrolyzable tannins is ellagitannins which can release ellagic acid by hydrolyzing in vivo and can produce urolithin through gut microbial metabolism. In general, the antimicrobial action of tannins has been determined in vitro, and the impact of these tannins on the abundance of the colonic ecosystem in vivo has not been sufficiently illustrated. 

Bialonska et al. (2009) used a liquid culturing method to investigate the influence of a 0.01% extract of commercial pomegranate and its major components at the level of 0.05% on the proliferation of different species of bacteria present in the human intestinal ecosystem in an in vitro study. The findings of this study demonstrated that byproducts of pomegranate and punicalagins suppressed the growth of pathogenic *S. aureus* and Clostridia, while ellagitannins normally did not affect the abundance of probiotic bifidobacteria and lactobacilli [189]. In another study, Bialonska et al. (2010) used a healthy individual’s fecal samples inoculated in a batch-culture fermentation system simulating the environment of the intestinal ecosystem to confirm the maintenance of the above trend. At the end of this study, pomegranate extract resulted in an accelerated increase in the total number of bacteria, promoting the growth of *Enterococcus*, *Bifidobacterium* spp., and *Lactobacillus*. At the same time, the abundance of the *C. histolyticum* group was not affected [190]. It was observed that pomegranate ellagitanins and urolithin A, which is their major metabolite produced by the microbiota, modified the composition of the colonic ecosystem in rats by increasing the prevalence of *Bifidobacterium* and *Lactobacillus* [191].

Moreover, Li et al. (2015) monitored alterations in the composition of GM in 20 healthy individuals consuming a pomegranate extract (POM) of 1 g for four weeks in an in vivo study. Individuals were classified into three separate groups depending on the amount of urolithin A in the feces and urine. In this experiment, it was observed that the abundance of beneficial Actinobacteria was significantly increased while those of *Firmicutes* in individuals producing urolithin A were reduced. Furthermore, in the fecal samples of individuals who produce urolithin A, the proportion of *Akkermansia muciniphila* of phyla Verrucomicrobia was greater than non-producers. After four weeks, the prevalence of *Lactobacillus*, *Escherichia*, *Butyrivibrio*, *Prevotella*, *Veillonella*, *Enterobacter*, and *Serratia* genera was enhanced while those of *Collinsella* in urolithin A producers was reduced. In some participants, POM extract ingestion led to the production of metabolites, which may stimulate beneficial health effects next to modification of the intestinal microflora [192].

Samanta et al. (2004) pioneered research conducted using an in vivo model to investigate the influence of tannic acid on the communities of culturable organisms. It was observed that tannic acid could alter the ecological balance of the intestinal microbiota of the rat. This study proved the toxic and anti-nutrient properties of tannic acid from the observations of enormous microbial growth and the reduction in the bodyweight of the experimental animal after tannic acid supplementation for 21 days [193]. Smith and Mackie (2004) stated that the tannin diet could reduce the abundance of low G + C Gram-positive bacteria. In contrast, the proportion of Enterobacteriaceae and *Prevotella*, *Bacteroides*, and *Porphyromonas* increased in the samples of rats [194].

##### Condensed Tannins (Proanthocyanidins)

Condensed tannins or proanthocyanidins are flavonoid oligomers that comprise one of two large groups of tannins. They are widely abundant in plants, including ferns, gymnosperm, and angiosperms and contribute to the purple, red, or blue colors of flowers, fruits, and sometimes leaves. In general, condensed tannins show antioxidant activity in their monomeric, oligomeric, or polymeric forms. Smith and Mackie (2004) investigated the effect of condensed tannins on the fecal microbial population of rats and found that dietary supplementation with condensed tannins extracted from *Acacia angustissima* modified the gut microbial population, resulting in a shift in the predominant bacteria towards tannin-resistant Gram-negative Enterobacteriaceae and Bacteroides species and a decrease in the abundance of the Gram-positive *C. leptum* group [194]. In another study, rats were fed proanthocyanidin-rich cocoa preparation, and there was a significant reduction in the abundance of *Clostridium*, *Bacteroides*, and *Staphylococcus* genera in the feces [195]. Further, apple proanthocyanidins showed a significant increase in the population of Actinobacteria when incubated with colonic microbiota in a batch culture model [196]. However, Tao et al. (2019) stated that the differences in the animal model and types and sources of proanthocyanidins affect the mutual relationship between proanthocyanidins and gut microbiota [197]. 

In terms of ruminant animal productivity and health, condensed tannins have many advantages and disadvantages, and these aspects will not be extensively addressed in this paper. In brief, condensed tannins can increase animal growth performance by increasing nitrogen retention in the diet, improve animal health via reducing the worm load in the intestine, and reduce enteric methane emission through their ability to lower the activity of methanogenic microbes in the rumen. 

### 4.2. Effect of Organosulfur Compounds

The organosulfur group comprises several compounds, including indoles, isothiocyanates, and allylic sulfur compounds. Generally, many cultures have used garlic (*Allium sativum* L.) in cooking and traditional medicine since ancient times. Garlic is the richest source of organosulfur compounds, containing about 1.1–3.5%. During the storage of intact garlic, S-allyl-l-cysteine sulfoxides (alliin) are produced by the hydrolyzation and oxidation of γ-glutamyl-S-allyl-l-cysteines (G-SAC), which is the chief organosulfur compound (OSC) in garlic [198]. Alliinase is released from garlic during chopping or slicing or crushing or chewing, by which alliin is activated to allicin and other thiosulfates [199]. Bioactive OSCs are responsible for several beneficial health impacts, mainly producing defense compounds, which have wide-ranging antimicrobial properties [200].

According to Zhai et al. (2018), the abundance of the family Lachnospiraceae was decreased by the alliin derived from garlic [201]. Chen et al. (2019) studied intact garlic’s impact and mode of action on the intestinal microflora using C57BL/6N male mice fed with or without whole garlic in a normal diet or a high-fat diet. It was found that ingestion of whole garlic, which constituted fructan, alliin, and other organosulfur derivatives, including allicin, S-allylcysteines and G-SAC could increase the α-diversity of the GM. This particularly enhanced the relative abundance of the Lachnospiraceae family and decreasing the population of *Prevotella* genus resulting in the attenuation of high-fat diet-induced dyslipidemia and disturbances of GM [202].

### 4.3. Effect of Carotenoids

Carotenoids are pigments that contribute to the red, orange, and yellow colors of several vegetables, fruits, and food products derived from them. Generally, carotenoids are classified into two groups: 1. carotenes (such as lycopene, α-carotene, and β-carotene) and 2. xanthophylls (such as zeaxanthin, lutein, and meso-zeaxanthin isomer) [203]. Humans must obtain carotenoids from the diet since they cannot synthesize carotenoids from endogenous precursors. The carotenoids in the blood have poor bioavailability (10–40%) as they are fat-soluble bioactive molecules and gut microbiota ferment the carotenoids once the enter the colon [204]. However, the biological roles of carotenoids in the intestinal ecosystem and their gut microbial utilization are yet to be fully understood [205]. Carotenoids act as antioxidants at low concentrations, while at high dosages, they have been shown in clinical trials to cause toxic effects based [206].

#### 4.3.1. Astaxanthin

Astaxanthin (Figure 15) is an oxycarotenoid pigment, and is widely distributed in certain marine animals, including shrimp and salmon as well as in particular microalgae [207]. In *Helicobacter pylori*-infected mice, the abundance of total bacteria was notably decreased, and gastrointestinal inflammation was reduced by the supplementation of astaxanthin [208]. In another study, astaxanthin-treated stressed rats showed a notable reduction in the abundance of microbial community and inflammation scores [209]. Later, Yonei et al. (2013) used a real-time PCR assay to evaluate the influence of astaxanthin on the alterations of the colonic microbiota expression in mice fed a high-fat diet (fat 35%). This study identified that the abundance of the Bacteroides genus, *C. leptum*, and *C. coccoides* species were increased. In contrast, the prevalence of the *Streptococcus* genus (lactobacilli) was reduced due to high-fat diet consumption compared with a control diet containing 3.9% of fat. However, the above modifications were inhibited in the mice receiving astaxanthin-supplemented diets [210].

Liu et al. (2018) used C57BL/6J mice as a biomedical model for humans to evaluate the intervention of astaxanthin on the gut microflora and to examine its ability to protect against liver injury induced by alcohol. The results showed that astaxanthin treatment notably increased the species from *Akkermansia* and *Verrucomicrobia* and reduced species from the genera *Parabacteroides*, *Bilophila*, and *Butyricimonas* and the phyla Proteobacteria and *Bacteroidetes* in comparison to the mice group fed with ethanol. Furthermore, astaxanthin supplementation considerably relieves inflammation and reduces the excessive accumulation of lipid and serum markers of liver injury [211]. The findings of an in vivo pilot study stated that intestinal microbiota could be altered by 0.04% (*w*/*w*) dietary astaxanthin at the phylum level by both genotype and gender [204]. Supplementation of astaxanthin selectively decreased the prevalence of colonic *Bacteroides* and Proteobacteria in female wild-type and BCO2 knockout C57BL/6J mice. Moreover, the population of *Bifidobacterium* and Actinobacteria significantly increased by astaxanthin only in male wild-type mice, which would result in a favorable effect on gut health.

#### 4.3.2. Lutein

Lutein (Figure 15) belongs to the xanthophyll group, and it is an oxygenated carotenoid pigment. Humans and mammals obtain lutein from their diet [212]. In a human phytotherapy study, two products which contained blackcurrant extract powder, lutein, and lactoferrin were identified to have potential prebiotic actions by remarkably enhancing the abundance of lactobacilli and bifidobacteria, while decreasing other bacterial loads, including *Clostridium* spp. and *Bacteroides* spp. [213]. Animals maintained under grazing or rangeland consume a larger amount of lutein over the lifespan than those grown indoors on formulated rations or feedlot diets. This lutein can be stored in adipose tissue and fat droplets within the muscle tissue of farm animals. It may act as a scavenger of free radical molecules or substances that reduce oxidative damage in the body [212].

#### 4.3.3. Lycopene

Lycopene (Figure 15), one of the primary carotenoids, is a red pigment widely distributed in watermelon, tomatoes, and some other fruits. Wiese et al. (2019) investigated the influence of dark chocolate and lycopene on intestinal microflora, skin, liver metabolism, blood, and oxygenation of skeletal muscle tissue. It was found that lycopene compounds showed alterations in the composition of colonic microflora by promoting the abundance of beneficial *B. longum* and *B. adolescentis*, and was dependent on the dosage level [214].

## 5. Mechanism of Action of Phytochemical

### 5.1. Effect on the Gut Microbiome

In general, dietary phytochemicals (polyphenols) are received as xenobiotics in humans once consumed. Their absorption in the small intestine and their biological availability is relatively low due to their structural complexity and polymerization [215]. Numerous studies revealed that dietary phytochemicals entering the intestinal ecosystem and their metabolites can alter the composition of microbial ecology beneficially by acting as prebiotics as well as antimicrobial agents against harmful gut microbiota [49,72,216]. The potential benefits of phytochemicals associated with GM are summarized in Figure 16. 

The structure of phytochemicals, the type of microbial strain, and the dosage level determine the effect of phytochemicals on bacterial growth, multiplication, and metabolism [217]. To exemplify, Gram-negative bacteria show greater resistance than Gram-positive bacteria to phytochemicals which may be due to the variations in the composition of their cell wall [218]. Numerous studies demonstrated several mechanisms of the effect of phytochemicals on bacteria cells, their growth, and propagation. Phytochemicals may alter the functions of the bacterial cell membranes and, thus, suppress the cell growth by binding to the membranes in a dose-dependent manner [219]. Through the production of hydrogen peroxide and via changing the permeability of the bacterial cell membranes, phytochemicals, including catechins, affect several species of bacteria such as *Klebsiella pneumonie*, *E. coli*, *Salmonella choleraesis*, *Bordetella bronchiseptica*, *Bacillus subtilis*, *Serratia marcescens*, *Pseudomonas aeruginosa*, and *S. aureus* [1,220].

Sirk et al. (2009) and Sirk et al. (2011) also stated that hydrogen bond formation between the hydroxyl groups and lipid bilayers of bacterial cell membranes may govern the anticancer, antimicrobial, and some other health benefits of theaflavins and catechins. The absorption of catechins and its ability to form a hydrogen bond with the groups of lipid heads are notably influenced by their molecular structure and collective conditions. The configuration of the theaflavins and catechins is affected by the molecular structure when binding to the surface of the lipid bilayer [221,222].

Stapleton et al. (2007) revealed that (−)-epicatechin gallate (ECg), which is a component of green tea, enhances the aggregation of staphylococcal cells, sensitizes methicillin-resistant *S. aureus* strains to β-lactam antibiotics, and increases the thickness of cell-walls. Variations in the physical properties of the bilayer, which is associated with ECg, can cause alterations in the structure of teichoic acid in the cell wall that leads to the modification of the features of the cell-surface required to support the phenotype, which is resistant to β-lactam [98]. 

Bacterial quorum sensing can also be influenced by phytochemicals via synthesizing, liberating, and detecting autoinducers, which are the small signaling molecules (for example, oligopeptides in Gram-positive bacteria and acylated homoserine lactones in Gram-negative bacteria) [223,224]. For instance, Hubert et al. (2003) stated that polyphenols influence the synthesis of small signaling molecules by bacteria such as *Burkholderia cepacia*, *Pseudomonas putida*, and *E. coli*, that stimulate an exponential increase in the abundance of bacteria [225]. In another study, Monagas et al. (2003) revealed that isolated or synthesized flavan-3-ols metabolites by Phase II-conjugation reactions have influence, rather than their antioxidant actions, by interfering with the signaling pathways involved in the process of disease development [226].

Additionally, the production of toxin VacA (Vacuolating toxin A), which is a primary virulence factor of *Helicobacter pylori*, was greatly suppressed by the polyphenols of green tea and red wine [227]. The destruction of bacteria cell membranes, suppression of urease activity, and disturbance of bacteria multiplication are some of the modes of inhibitory action of dietary phytochemicals on *H. pylori*. Through these mechanisms, cells become more sensitive to foreign substances such as antibiotics, resulting in the disturbance of the proton motive force via functions related to the cell membrane and the loss of H^+^-ATPase [228].

Furthermore, the suppressive action on the synthesis of DNA and RNA is possibly due to the influence of the flavonoids’ B-ring on the hydrogen bonding of the flavonoids with nucleic acid bases [229]. Plaper et al. (2003) stated that quercetin impedes the ATPase activity of the enzyme by binding to the GyrB subunit of *E. coli* DNA gyrase [230]. To confirm previous conclusions, Gradisar et al. (2007) revealed that catechins can attach to the ATP binding site of the gyrase B subunit and thus suppress the DNA gyrase of bacteria [231].

In vitro and animal experiments have shown that polyphenol compounds might suppress the generation of water-insoluble glucans, and this might be the reason for the anticaries action of cocoa powder [232,233]. Moreover, onion extract, a rich source of flavonoids, has been revealed to affect important bacteria which cause adult periodontitis, such as *Prevotella intermedia* and *Porphyromonas gingivalis*, and also on *Streptococcus sobrinus* and *Streptococcus mutans*, which also contribute to harmful effects on gut health [234].

A further study suggested that sensitive gut bacterial populations, mostly aerobic microbes, might be affected by iron deficiency in the gut due to the formation of polyphenol-metal ion complexes [235]. Iron is necessary for aerobic micro-organisms for different functions, such as forming heme groups and decreasing DNA ribonucleotide precursors. However, Freestone et al. (2007) suggested that the abundance of enteropathogenic bacteria may be increased by dietary catechols, which supply iron under conditions when the iron is limited and allow the growth of intestinal bacteria [236]. Further research in animals and humans is necessary to understand different mechanisms and mode of actions of phytochemicals on particular gut microflora proliferation and functions since most of them are not fully clarified at this time. 

### 5.2. Studies Performed on the Gut Microbiome of Animals

Health and medical studies aimed at human intervention trials are normally conducted in phases, with animal studies a first step prior to human trials. Biological effects and mechanisms of action that occur in the human body could be better understood by working with animals as a biomedical model for humans. The main focus of animal studies has been the evaluation of the safety of the compound and understanding the metabolism of phytochemicals, particularly their influence on microbial action, digestibility and metabolic diseases in humans, and animals that are used for human purposes such as food, companion, clothing, travel, recreation, etc. The interaction between the composition and diversity of the gut microflora and the metabolites of phytochemicals that are derived from the host has been investigated in only a small number of animal studies. Culture-independent comparison studies have revealed that even though the distal portion of the intestinal ecosystem of humans and mice inhabits similar phyla of bacteria, many species and genera of bacteria present in mice are not found in humans. Therefore, when generalizing the findings of the animal studies to humans, caution should be taken [237,238]. Studies carried out in animals to evaluate the impacts of phytochemicals on the alteration in the composition of the gut microbiome are summarized in Table 1.

Although studies have been widely carried out in rats, some experiments used larger animals, including pigs, cattle, chickens, or sheep. Animal experiments conducted on calves [241] and pigs [239] demonstrated that the gut microbial composition was improved by the supplementation of tea polyphenols. In a study using pigs, the supplementation of tea polyphenols notably enhanced the growth of lactobacilli while decreasing the abundance of Bacteroidaceae and total bacteria. Furthermore, there was a tendency to reduce clostridia that was lecithinase-positive such as *C. perfringens* [239], though, *Lactobacillus* spp. and *Bifidobacterium* spp. showed a slow rate of the decrease, and *C. perfringens* showed a faster reduction rate in calves administrated with the extract of green tea [241]. 

Kafantaris et al. (2017) investigated the effect of grape pomace, a by-product of the wine-making process rich in polyphenols, on the gut microbiota of lamb. Twenty-four lambs were divided into two experimental groups, one receiving control feed and the other receiving a diet supplemented with grape pomace for 55 days. The effect was examined by analyzing the population of fecal microflora of the lambs. It was found that the experimental feed suppressed the growth of pathogenic bacteria *E. coli* and *Enterobacteriacae* while inducing the growth of facultative probiotic bacteria [242]. In another study, the influence of commercial seaweed extract, collected in northern Norway, on the intestinal microflora of Norwegian white sheep ewes was evaluated in vivo and in vitro. The lactic acid bacteria count in the ewes was decreased and the growth of *Enterococcus* sp. was inhibited by the bioactive compounds present in the extracts of red and brown seaweeds [243].

Ruminants (cattle, sheep, goats, alpacas, deer, etc.) cannot directly digest plant material due to the lack of enzymes capable of breaking down cell walls (cellulose and hemicellulose). Rather, the GI of ruminants is inhabited by a large population of bacteria, protozoans, and fungi in their four-chambered stomach (rumen, reticulum, omasum, and abomasum) capable of digesting a diet with a large proportion (80–85%) of high-fibrous plant materials (roughage diets), whereas monogastric animals consume smaller proportion (15–20%) of roughage diets. The micro-organisms present play a major role in the degradation of undigestible fibrous materials enabling the use of nutrients by themselves as well as providing a medium for digestion and absorption in the small (duodenum, jejunum, and ileum) and large (cecum, colon, and rectum) intestines of the host animals. The rumen and reticulum are home to the micro-organisms involved in the fermentation and breakdown of plant materials, producing volatile fatty acids and releasing other nutrients that both microbes and host animals use for energy and stimulating several metabolic reactions. Partially or fully digested food passes through the last compartment of the stomach (abomasum) to the small intestine and large intestine, and further digestion and absorption of nutrients take place by the host animal. The anatomical and functional attributes of the small intestine of ruminants is similar to non-ruminants and ranges in length by approximately 12–30 times the animal’s body length [6].

The microbial community (bacteria, protozoa, archaea, fungi, and bacteriophage) in the rumen is responsible for the fermentation of complex plant materials consumed by ruminants [252,253]. However, methane gas, a greenhouse gas with global warming potential, is generated as a metabolic by-product of enteric fermentation by rumen methanogens [254]. Studies conducted in ruminants show that dietary supplementation with phytochemicals could reduce methane gas emission by modulating the rumen microbiota. For instance, over 21 seaweeds have been proven to decrease methane emission in vitro [255] which is believed to be, at least partially, due to the bio-actives and/or phytochemicals present in seaweed. The production of methane gas by ruminants accounts for nearly 81% of greenhouse gas emissions from the livestock sector, 90% of which comes from rumen microbial methanogenesis [256,257,258]. 

A complex community of ciliate protozoan, anaerobic fungi, and bacteria produce H_2_ and CO_2_ in the rumen. These gases are converted to methane by a population of methanogenic archaea [259,260]. Feeding ruminants with diets containing phytochemicals could reduce the population of methanogenic archaea, ciliate protozoan, anaerobic fungi, and bacteria in the rumen, thereby reducing methane gas emissions. For example, saponin compounds, which are defaunation agents for protozoa, could reduce methane production by decreasing the abundance of protozoa and their associated methanogens in rumen fluid [261]. In the future, climate variability and the sustainability of animal production are challenging areas in managing the ecosystems in relation to methane mitigation. Selecting multispecies pastures and fodders containing different types and levels of phytonutrients, or by supplementing agricultural by-products consisting of bioactives/phytochemicals to ruminants could potentially improve milk and meat productivity while reducing the enteric methane emission [262]. It is feasible to manipulate the activity of methanogenic bacteria and protozoa in the rumen (gut microbiota) resulting in lower enteric methane emission and improved productivity. Some examples are, the inclusion of seaweeds, oilseed by-products, oils, and by-products of the wine industry into ruminant diets has merits in reducing the activity and/or proliferation of methanogenic archaea in the rumen and diverting this dietary energy loss as methane emission into body gain [263,264,265]. 

The findings of studies using rats demonstrated that a significant modification in the intestinal microflora was observed in carcinogen-treated F344 rats supplemented with wine polyphenols in comparison to rats fed control diets [244]. Even though the feces of polyphenol-supplemented rats and control rats had the equivalent total number of bacteria and ratio between aerobic and anaerobic micro-organisms, the abundance of bifidobacteria and lactobacilli was enhanced and those of Clostridia, *Propionibacterium*, and *Bacteroides* was reduced. The authors suggested that the influence of wine phytochemicals on the function of the gut and carcinogenesis may be associated with alterations to the intestinal ecosystem, decreases in oxidative damage, and changes in gene expression.

In another study, fresh stool samples of rats that were administrated with apple juice as a replacement for drinking water, exhibited a higher abundance of bifidobacteria and lactobacilli, which varied from the rats fed a control diet by one-log10 colony-forming units [266]. Molan et al. (2010) further proved the prebiotic properties of extracts of wild blackcurrant in rats following the findings of previous in vitro experiments. After regular supplementation of those extracts to rats, a notable growth in bifidobacteria and lactobacilli was recorded [267]. In an earlier study, the influence of the supplementation of extracts of grape pomace on the modification of the gut microbiome of broiler chicks was investigated [250]. The results demonstrated that the prevalence of *Enterococcus*, *E. coli*, and *Lactobacillus* species was higher in the gut of birds fed with grape extracts than in other birds. In conclusion, it was stated that the intestinal morphology and gut microflora were altered by the compounds rich in polyphenols from grapes, which improved the extent of the biodiversity of gut microbiota in broiler chicks.

Overall, numerous studies suggested that the application of the appropriate animal model provides a robust technique to explore the gut microbial diversity and associated metabolic functions, even though the intestinal microflora of animals and humans is not entirely the same. In general, the presence/absence and abundance of a gene, profiles of the gut microbial ecosystem, and the range of functions of microbes can be determined by metagenomic studies. However, they can influence only an observed phenotype because the presence of a gene does not indicate its functionality or that expression is there [268].

## 6. Gut Microbiota and Metabolic Diseases (MD)

### 6.1. Influence of Gut Microbiota on some Metabolic Diseases

Socio-demographic and environmental factors significantly contribute to the status of people with metabolic disorders, including diabetes mellitus, whereas human genetics show a lower influence [269]. Colonic microflora also shows a constructive interaction with metabolic diseases. In addition to its digestive functions, the intestinal ecosystem maintains the optimum condition of human health via contributing non-human genome encoded enzymes. This includes the generation of vitamins and the breakdown of polyphenols and polysaccharides [59]. The intestinal microflora is also responsible for the etiologies of metabolic disorders, such as obesity, hypertension, cardiovascular disease, diabetes mellitus, and inflammation.

It is reported that intestinal microflora controls most of the features of human physiology, including regulation of colonic function, immune system modification, exogenous toxins removal from the body, and defense mechanism against several pathogens. Several epidemiological and experimental data revealed that energy homeostasis and maladaptation are significantly influenced by intestinal microbiota, which is interrelated with insulin resistance and obesity [270,271].

Furthermore, Clarke et al. (2010) stated that metabolic disorders may be regulated by intestinal microbiota via affecting the immune system of the host and modifying the inflammatory signaling pathway [272]. It was found that the incidence of various features of metabolic disorders can be induced and suppressed by the lack of Toll-like receptors functions, namely TLR4 and TLR5 [273,274]. These toll-like receptors can recognize the pattern of the cell membrane, and they significantly contribute to the non-specific defense mechanisms of the host [274]. In an earlier study, Clarke et al. (2010) indicated that intestinal microflora also can produce other pro-inflammatory compounds, including flagellin, peptidoglycan, and lipoproteins, etc., which can bind to toll-like receptors [272]. Overall, the colonic microbiota plays a major role in the incidence of metabolic disorders via altering the signaling pathways associated with the initiation and progression of inflammations. 

### 6.2. Impact of Phytochemicals on Metabolic Syndrome by Modulating Gut Microbiota

It was observed that chronic low-grade inflammation is generated due to the complex interaction between a person’s diet and the intestinal microbiota [275,276]. The key reason for indicating that metabolic syndrome is inter-related with chronic diseases is the reciprocal association between the diet and the gut microbiota. 

Several researchers focused on phytochemicals because of their significant influence on human health. Even though the presence of dietary fibers, vitamins, minerals, etc. are responsible for the beneficial health effects of fruits, many studies revealed that the phytochemical content of the fruits also favorably influences human health by decreasing the complications related to obesity and other disorders [171,277,278]. Numerous experimental and epidemiological data demonstrate that phytochemicals can protect against diabetes and its related complications by modifying intestinal microbiota [279,280]. Figure 17 shows a simple illustration of the impact of phytochemicals and a diet rich in phytochemicals on metabolic diseases by modifying the intestinal microflora.

Gut dysbiosis can be inhibited with the supplementation of phytochemicals, which enhances the prevalence of several species of beneficial bacteria and β-diversity of microflora while reducing the abundance of opportunistic harmful bacteria. Alterations in colonic microflora can lead to enhanced intestinal barrier function, increased breakdown of glycan, and expenditure of energy. It can alleviate the incidence of adiposity, inflammation, dyslipidemia, weight gain, and insulin resistance. Abovementioned beneficial alterations in the host body consequently resulted in decreased metabolic diseases and its related complications [53]. 

## 7. Conclusions and Future Perspectives

In recent years, the interaction of dietary phytochemicals with the intestinal microflora has garnered a greater interest because of their impact on gastroenterology, fermentation, digestion, disease prevention, and human health. Several preclinical and clinical studies revealed that phytochemicals could act as antimicrobial agents and exhibit prebiotic effects on harmful intestinal microbiota. Despite numerous studies on this topic, the clarification and understanding of the specific mechanism of each phytochemical are not well understood. Complexities during the implementation, complications in interpreting the in vitro findings into in vivo applications, and ethical and economic issues are some constraints for in vivo experiments. The review indicates that phytochemicals play a crucial role in improving human (and animal) health, particularly gut health, via promoting the abundance of favorable bacteria and protozoa and suppressing harmful bacteria, exhibiting prebiotic activities. The bioactivity and metabolic function of some phytochemicals in the body are examined so far. Therefore, further studies should focus on examining the therapeutic potential of remaining phytochemicals and explaining the complex mechanisms and identifying which phytochemical will specifically influence which micro-organism in modulating gut microbiota and maintaining good health. Furthermore, the interaction of dietary phytochemicals with other nutrients, such as minerals, vitamins and essential fatty acids, in the microbiome and their effects in terms of fermentation, digestion, bio-accessibility of nutrients in the gut, and the maintenance of health in humans and animals using in vivo studies warrants further research. Due to inadequate human trials and animal model studies, another challenge is to utilize these natural phytochemicals in the pharmaceutical industry to develop drugs targeting improved gut health. In future, more animal model studies and human trials should be conducted to pave the path for drug development using phytochemicals alone or diets containing phytochemicals and other nutrients such as vitamins, essential fatty acids, etc.

## Figures and Tables

**Figure 1 ijms-23-08124-f001:**
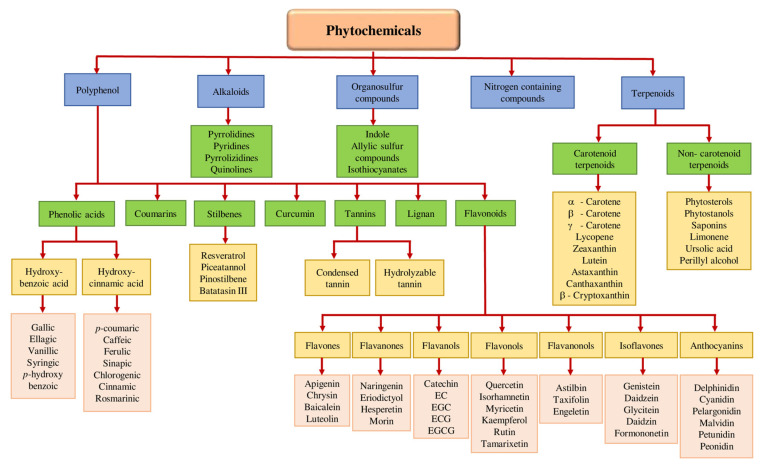
Classification of Dietary Phytochemicals (self-generated) [EC: epicatechin; EGC: epigallocatechin; ECG: epicatechin gallate; EGCG: epigallocatechin 3-gallate].

**Figure 2 ijms-23-08124-f002:**
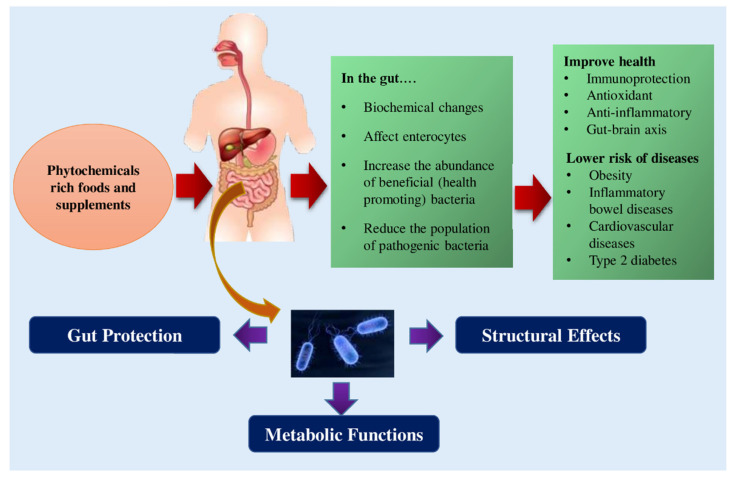
The influence of phytochemicals on GM and key roles of GM in humans (Self-generated).

**Figure 3 ijms-23-08124-f003:**
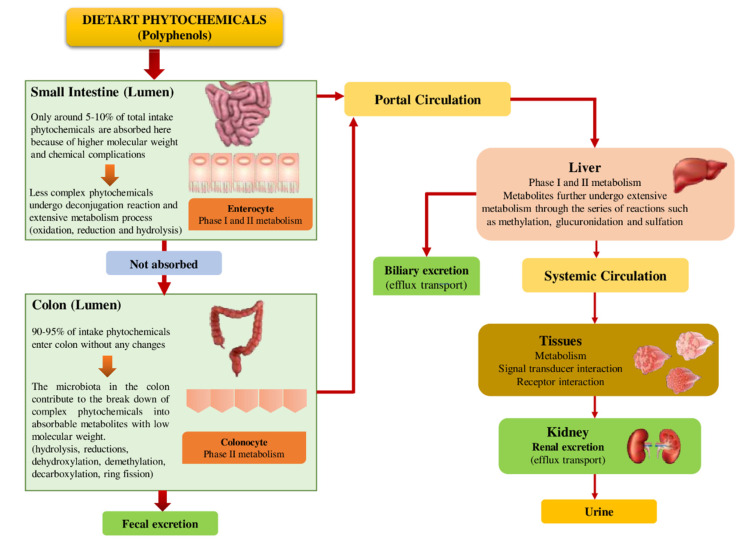
Simple illustration of the metabolic pathway of dietary phytochemicals in human body (self-generated).

**Figure 4 ijms-23-08124-f004:**
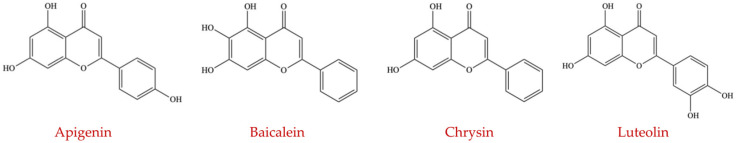
Chemical structures of flavones (apigenin, baicalein, chrysin, and luteolin) (self-generated).

**Figure 5 ijms-23-08124-f005:**
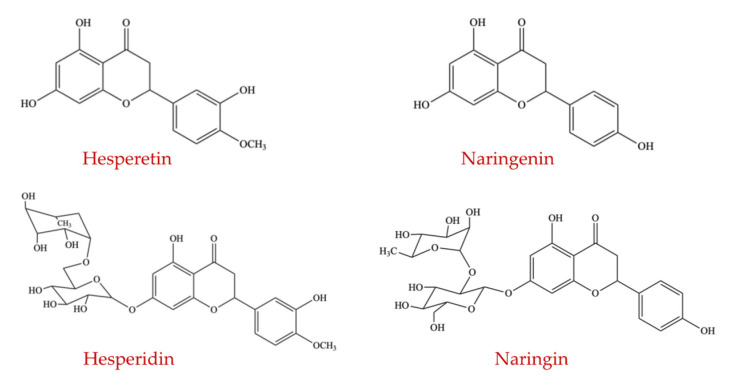
Chemical structures of flavanones (hesperetin, naringenin, hesperidin, and naringin) (self-generated).

**Figure 6 ijms-23-08124-f006:**
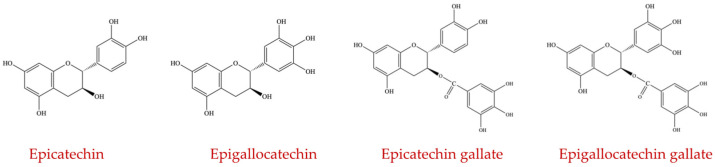
Chemical structures of flavanols (epicatechin, epigallocatechin, epicatechin gallate, and epigallocatechin gallate) (self-generated).

**Figure 7 ijms-23-08124-f007:**
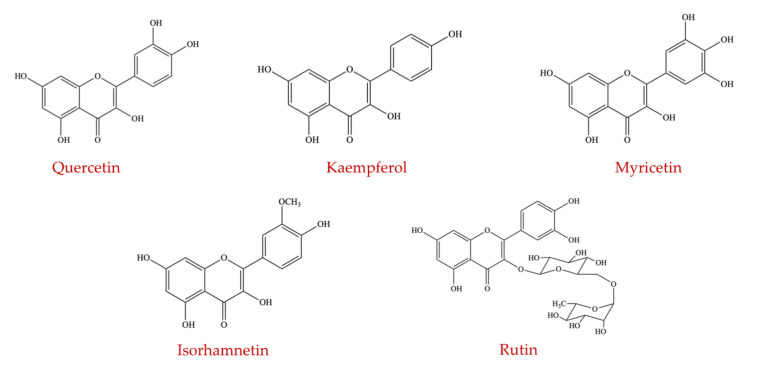
Chemical structures of flavonols (quercetin, kaempferol, myricetin, isorhamnetin, and rutin) (self-generated).

**Figure 8 ijms-23-08124-f008:**
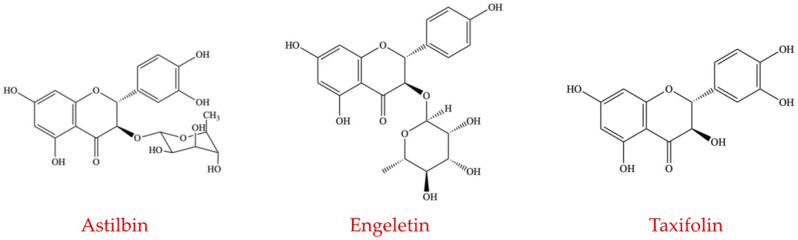
Chemical structures of flavanonols (astilbin, engeletin, and taxifolin) (self-generated).

**Figure 9 ijms-23-08124-f009:**
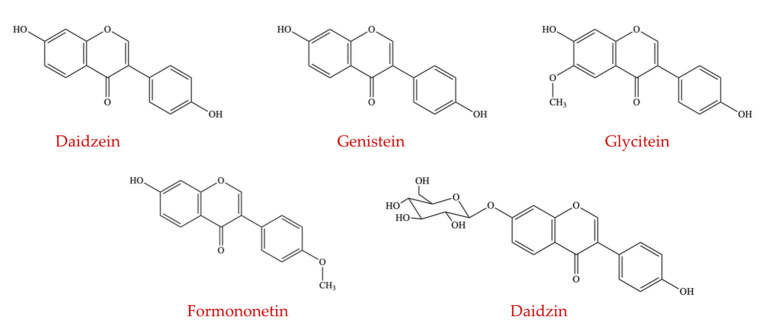
Chemical structures of isoflavones (daidzein, genistein, glycitein, formononetin, and daidzin) (self-generated).

**Figure 10 ijms-23-08124-f010:**
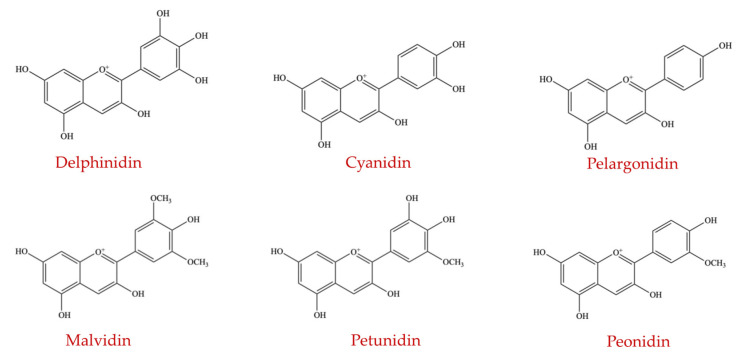
Chemical structures of anthocyanins (delphinidin, cyanidin, pelargonidin, malvidin, petunidin, and peonidin) (self-generated).

**Figure 11 ijms-23-08124-f011:**
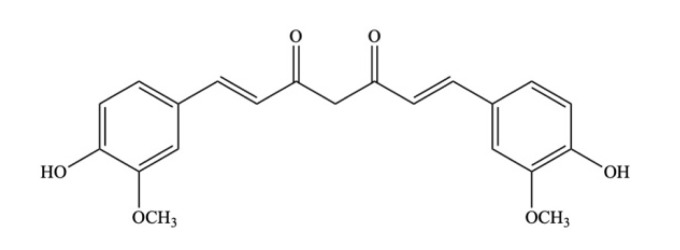
Chemical structure of curcumin (self-generated).

**Figure 12 ijms-23-08124-f012:**
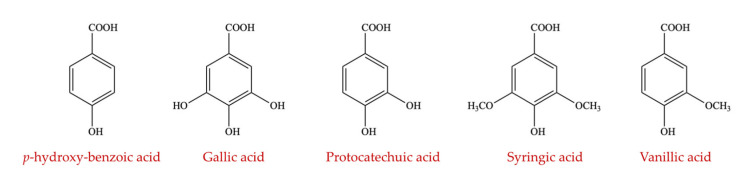
Chemical structures of hydroxybenzoic acids (*p*-hydroxybenzoic acid, gallic acid, protocatechuic acid, syringic acid, and vanillic acid) (self-generated).

**Figure 13 ijms-23-08124-f013:**
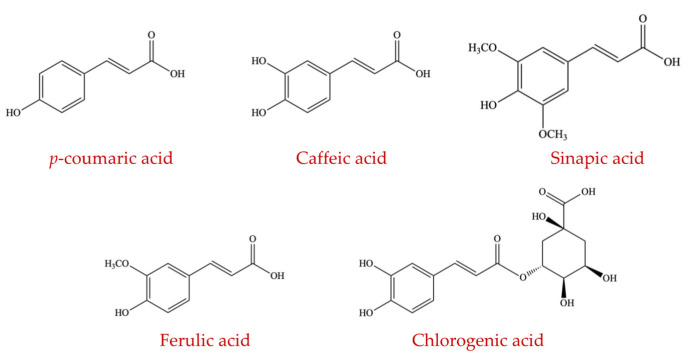
Chemical structures of hydroxycinnamic acids (*p*-coumaric acid, caffeic acid, sinapic acid, ferulic acid, and chlorogenic acid) (self-generated).

**Figure 14 ijms-23-08124-f014:**
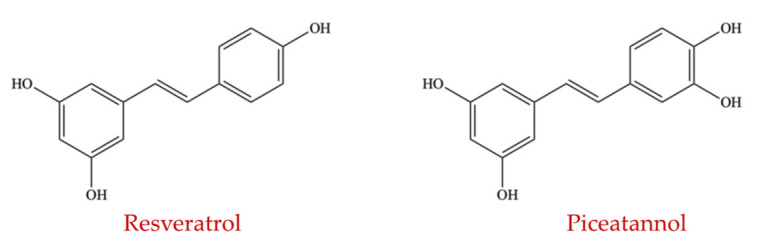
Chemical structures of resveratrol and piceatannol (self-generated).

**Figure 15 ijms-23-08124-f015:**
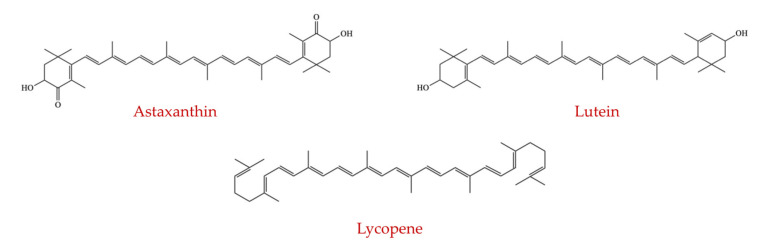
Chemical structures of carotenoids (astaxanthin, lutein, and lycopene) (self-generated).

**Figure 16 ijms-23-08124-f016:**
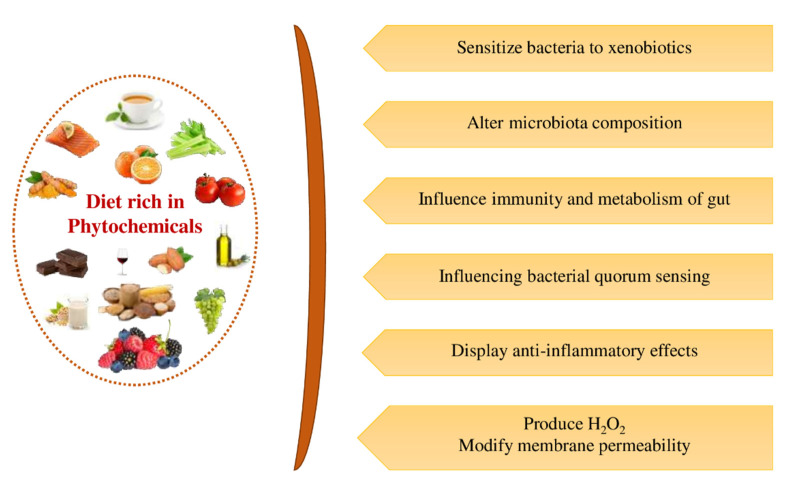
The potential benefits of phytochemicals associated with gut microbiota (self-generated).

**Figure 17 ijms-23-08124-f017:**
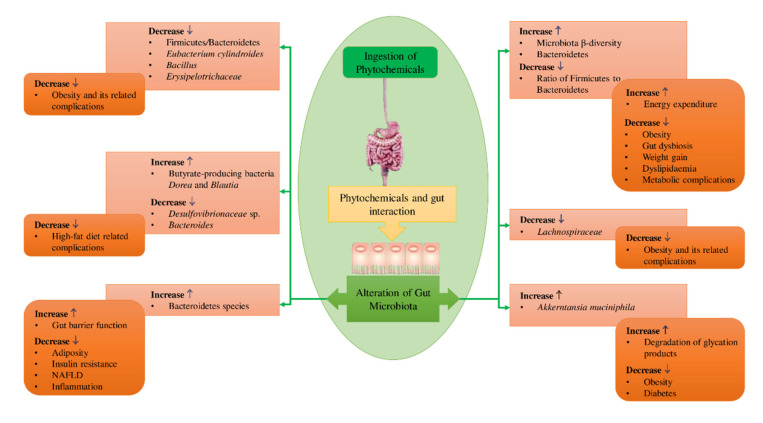
Impact of phytochemicals and diet rich in phytochemicals on metabolic diseases by modifying intestinal microflora (self-generated).

**Table 1 ijms-23-08124-t001:** Studies carried out in animals to evaluate the impacts of phytochemicals on the alteration in the gut microbiome.

Phytochemical	Animal	Effect on Microbiome and Related Mechanism	Reference
Tea polyphenols	Pigs	Enhanced the prevalence of lactobacilli, while reducing that of Bacteroidaceae, *C. perfringens*, and total bacteria	[239]
Coffee and caffeic acid	Rats with colon cancer	Supplementation specifically suppressed neoplastic cell transformation and colon cancer metastasis in mice via inhibition of TOPK (T-LAK cell-originated protein kinase) and MEK1	[240]
Green tea extracts	Calves	Decreased the abundance of *C. perfringens*, *Bifidobacterium* spp., and *Lactobacillus* spp.	[241]
Grape pomace extracts	Lamb	Suppressed the growth of pathogenic bacteria *E. coli* and *Enterobacteriacae* while inducing the growth of facultative probiotic bacteria	[242]
Seaweed extract	White sheep ewes	Lactic acid bacteria count in ewes and lambs was decreased and the growth of *Enterococcus* sp. was inhibited	[243]
Red wine extract rich in proanthocyanidin	Rats with colon cancer	Supplemented rats showed a significantly greater abundance of *Bifidobacterium* spp., *Bacteroides*, and *Lactobacillus* and a reduced prevalence of *Clostridium* spp.	[244]
Quercetin	High-fat-diet fed rats	Down-regulated *Eubacterium cylindroides*, Erysipelotrichaceae, and *Bacillus*. Decreased body weight. Reduced the abundance of *Bacillus* genus, *Firmicutes*, and Erysipelotrichi class.	[104]
Proanthocyanidins extracted from ***Acacia*** ***angustissima***	Rats	Increased the prevalence of *Porphyromonas* group, *Bacteroides fragilis* group, Enterobacteriaceae, and Bacteroides *Prevotella* and reduced the abundance of *C. leptum* group	[194]
Resveratrol	Rats with DSS-induced colitis	Promoted the cell counts of *Bifidobacterium* spp. and *Lactobacillus* in feces	[245]
Polyphenols present in Chinese propolis, Brazilian propolis	Rats with DSS-induced colitis	Altered the composition of intestinal microflora, including a decrease in *Bacteroides* spp.	[246]
Lowbush wild blueberries	Rats	Increased the population of *Slackia* spp., *Thermonospora* spp., and *Corynebacteria* spp., while reducing that of *Enterococcus* spp. and *Lactobacillus* spp.	[247]
Resveratrol	Rats with colon cancer	Decreased functions of host intestinal mucosal and fecal enzymes, including β-galactosidase, α-glucoronidase, α-glucosidase, nitroreductase, and mucinase	[248]
Polyphenols from fungi	Rats with DSS-induced colitis	Modified the composition of colonic microflora by decreasing the ratio of *Bacteroidetes* to *Firmicutes* and restoring the abundance of *Lactobacillus* spp.	[249]
Grape pomace concentrate (GPC), grape seed extract (GSE)	Broiler chicks	Increased the population of *Lactobacillus* spp., *E. coli*, and *Enterococcus* spp.	[250]
Polyphenols present in *Prunella* *vulgaris* honey	Rats with DSS induced colitis	Modified the composition of colonic microflora, by increasing the ratio of *Bacteroidetes* to *Firmicutes* and restoring the abundance of *Lactobacillus* spp.	[251]

## Data Availability

This is a review article. This study did not report any data.

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
