# Peer review of "The Impact of Plant Phytochemicals on the Gut Microbiota of Humans for a Balanced Life"

_ijms, 2022, doi:10.3390/ijms23158124_

Round 1
Reviewer 1 Report
In this work, the authors evaluate the impact of plant phytochemicals on the gut microbiota of human for balanced life. The authors have utilized appropriate technique for the selection of the article and, on the whole, the manuscript can be of interest for the reader of the journal. Some sentence are rambling and some issue must be addressed in the work. The changes are listed before:
- In the figure 1 plesase delete flavonones. Hesperidin and naringin are flavanones and bacalein is a flavone.
- Delete also the corresponding paragraph to flavonones it is not a real flavonoids subclasse, and introduces the corresponding part in the right subclasses.
- The work shows several of typewrite errors (such as m2) please change along all the work
- In my opinion is suitable to introduce a other figure to increase the reader interest
- The conclusion must be implemented to better explain the potentiality of the review and the future perspective
Reviewer 2 Report
The paper of Santhiravel et al. “The Impact of Plant Phytochemicals on the Gut Microbiota of Human for Balanced Life” aimed to reviewing of effects of phytochemicals on the modulation of the gut microbiota environment. I have some notes on this review that need to be corrected.
Overall text: the structure of the mentioned compounds should be included in the text.
4.1.1.3: the misspelling “flavonones” should be “flavones”; the section should be combined with 4.1.1.1.
4.1.1.3: hesperidin and naringin are flavanones not flavones; both a glycosides of flavanone aglycones hesperitin and naringenin, respectively;
Line 409 (and overall text): gamma-valerolactone can be replaced by γ-valerolactone (also alfa α, etc).
Line 418 (and overall text): the O symbol (for example, 3-O-(3-O-methyl) gallate) needs italic.
Line 452 (and overall text): the trans needs italic.
4.1.1.6: the misspelling “flavononol” should be “flavanonol”.
4.1.1.6: genistein is not flavanonol; it is isoflavone.
Line 520 (and overall text): bifidobacteria and lactobacilli are not Latin names (unlike Bifidobacterium and Lactobacillus) therefore, there are no need to use italic.
Line 529: Firmicutes needs italic.
Line 541: not everyone antocyanins are water-soluble.
Line 543: anthocyanins occur in nature in aglycone and glycoside forms.
Line 555: the bad idea to use condensed version B. infantis, B. bifidum, L. acidophilus, and B. adolescentis in review, it's confusion.
Line 564 (and overall text): the type of glycosides should be marked as O- or C-; for example, malvidin-3-glucoside should be malvidin-3-O-glucoside.
Reviewer 3 Report
The extensive review article presented by the authors addresses a recent topic that should continue to be extensively explored in the future. A very thorough work was carried out by the authors, who managed to compile a lot of relevant scientific information regarding the effects of phytochemicals on the gut . In my view, the review presented has total relevance to science and for healthcare professionals. In addition, the article is very well structured, organized, with clear and objective information, the authors are to be congratulated for the excellent work done. In my opinion, the article is ready for publication after the small minor reviews
Line 146 – correct m2 to m²
In figure 2, write the meaning of the abbreviation SCFs. Change the description on the balloon (phytochemicals (polyphenols), phytochemical rich foods and supplements). In my opinion it is better to remove “phytochemicals(polyphenols)”
Line 614. Curcumin is a polyphenol and not a subgroup of phenolic acids, review this part of the article.
LINE 658, correct HCA to HCAs
Line 679, I recommend dropping the term “recent study”, this information will not make sense 3-5-10 years from now. Do the same with other time-related terms throughout the manuscript.
Round 2
Reviewer 2 Report
The revised version of Santhiravel and co-author's manuscript looks great and no need any scientific correction. It may be accepted to publication in present form. Good luck.